# Ageing impairs the regenerative capacity of regulatory T cells in mouse central nervous system remyelination

Alerie Guzman de la Fuente [1,2,3] ✉, Marie Dittmer[1], Elise J. Heesbeen[1,12], Nira de la Vega Gallardo[1], Jessica A. White [1], Andrew Young[1,13], Tiree McColgan [1], Amy Dashwood[4,5], Katie Mayne[1], Sonia Cabeza-Fernández[2,3], John Falconer[1,14], Francisco Javier Rodriguez-Baena[3], Christopher E. McMurran [6,7], Mohammed Inayatullah[8,9], Khalil S. Rawji[10], Robin J. M. Franklin[10], James Dooley[4], Adrian Liston [4], Rebecca J. Ingram[1], Vijay K. Tiwari [1,8,9,11], Rosana Penalva[1], Yvonne Dombrowski [1] & Denise C. Fitzgerald [1] ✉

Myelin regeneration (remyelination) is essential to prevent neurodegeneration in demyelinating diseases such as Multiple Sclerosis, however, its efficiency declines with age. Regulatory T cells (Treg) recently emerged as critical players in tissue regeneration, including remyelination. However, the effect of ageing on Treg-mediated regenerative processes is poorly understood. Here, we show that expansion of aged Treg does not rescue age-associated remyelination impairment due to an intrinsically diminished capacity of aged Treg to promote oligodendrocyte differentiation and myelination in male and female mice. This decline in regenerative Treg functions can be rescued by a young environment. We identified Melanoma Cell Adhesion Molecule 1 (MCAM1) and Integrin alpha 2 (ITGA2) as candidates of Treg-mediated oligodendrocyte differentiation that decrease with age. Our findings demonstrate that ageing limits the neuroregenerative capacity of Treg, likely limiting their remyelinating therapeutic potential in aged patients, and describe two mechanisms implicated in Treg-driven remyelination that may be targetable to overcome this limitation.

Myelin is critical to the metabolic support and function of axons in the central nervous system (CNS)[1]. CNS myelin is produced by oligodendrocytes and damage to oligodendrocytes or myelin causes neurological impairment[2–4]. Myelin regeneration (remyelination) is highly neuroprotective and holds potential to restore lost function for patients with demyelinating diseases such as Multiple Sclerosis (MS)[5–8]. Like most regenerative processes, remyelination efficiency declines with age[9–11], contributing to age-associated disease progression and the accumulation of irreversible disability[12]. Age-associated remyelination decline has been linked to intrinsic changes in the capacity of

oligodendrocyte progenitor cells (OPC) to respond to damage in the aged CNS[13–15], changes in physical properties and the microenvironment (e.g extracellular matrix) of the aged CNS which makes it refractory to remyelination[16,17] and impaired capacity of other cells involved in the process of myelin regeneration (e.g., macrophages, microglia, astrocytes) to drive this process[18–23]. Despite the critical neuroprotective role of remyelination[8,24], there are currently no approved remyelination-enhancing therapies, a key unmet medical need. Thus, understanding how ageing alters the different cellular mechanisms governing myelin regeneration is key to overcoming age-

associated remyelination failure. This holds potential to prevent neuronal loss and the accumulation of irreversible cognitive, sensory and motor disabilities.

Regulatory T cells (Treg) have recently emerged as key cellular players in tissue regeneration due to both anti-inflammatory functions and capacity to modulate tissue-resident stem cell differentiation[25,26]. Similar crosstalk has also been described in the CNS, where we and others showed that Treg promote OPC differentiation and CNS remyelination[27–29]. The discovery of pro-remyelinating functions of Treg opened a new therapeutic avenue to prevent remyelination decline and associated neurodegeneration. However, whether this important neuroregenerative function of Treg is impaired with age remains unknown. Understanding how ageing affects myelin-regenerative functions of Treg is essential to determine the relevance and suitability of potential Treg-based pro-remyelination therapies in patients of advanced age in MS and other demyelinating diseases.

Here, we show that ageing impairs the capacity of natural Treg to drive OPC differentiation in vitro and myelination ex vivo. Surprisingly, this defect was reversible in a young remyelinating environment in vivo. Transcriptomic comparison of young and aged Treg identified Melanoma Cell Adhesion Molecule 1 (MCAM1) and Integrin alpha 2 (ITGA2) as candidates of Treg-mediated OPC differentiation that decrease with age. This study highlights Treg-driven OPC differentiation as a remyelination mechanism that is impaired by ageing and identifies Treg-expressed candidates (ITGA2, MCAM), which enhance OPC differentiation and should be further investigated to determine its potential relevance as therapeutic targets to promote CNS remyelination.

## Results

### Tissue-resident Treg are increased in the aged CNS

We first examined the frequency of murine Treg with age, and in agreement with previous studies[30,31], found increased numbers and proportions of Treg in aged mice compared to young mice in spleen and blood (Fig. 1A–E). Next, we investigated whether ageing impacted T cell populations in the CNS. As described previously[32–34], we observed an increase in the proportion of CD8[+] and CD4[+] T cells in aged murine CNS tissue with significant differences observed more prominently in spinal cord than brain (Supplementary Fig. 1A–H). We also observed a significant increase in the proportion of Treg in the spinal cord (Fig. 1F, G) and a non-significant trend in the brain (Fig. 1I, J). Proportions of tissue-resident (CD69[+]) Treg[35] were increased both in the aged brain and spinal cord (Fig. 1H, K). Previous studies showed that ageing impairs migration of Treg to damaged muscle[36]. To determine if ageing affects the capacity of Treg to accumulate in demyelinated lesions in the CNS we used two models of demyelination, lysolecithin-induced demyelination, and experimental autoimmune encephalomyelitis (EAE), a model of immune-mediated demyelination. Despite the low frequency of Treg in demyelinated areas, we did not observe any differences in Treg density in either of the two models at the time points investigated (Fig. 1L-–N). This suggests that aged Treg have a similar capacity to accumulate in demyelinated CNS tissue when compared to young Treg although differences at other phases of CNS damage cannot be conclusively ruled out.

### Expansion of Treg does not rescue impaired OPC differentiation in aged mice

We next verified that oligodendrocytes numbers were comparable in the young and aged healthy CNS (Supplementary Fig. 2A–D)[37,38], but that OPC differentiation into CC1[+] oligodendrocytes was impaired in aged demyelinated lesions in vivo as previously reported[10,13] (Supplementary Fig. 2E, F). We next investigated whether expanding Treg populations in aged mice would recover oligodendrocyte differentiation. Administration of IL-2/anti-IL-2 in vivo significantly expanded systemic CD25[+]Foxp3[+] Treg (~3-5-fold increase in blood and spleen) (Fig. 2A–C) with a trend of higher densities of Treg in demyelinated

lesions of Treg-expanded mice (Supplementary Fig. 2G–J) which was not statistically significant although numbers were low. We first examined whether Treg expansion enhanced the remyelination response in young mice. As myelin regeneration is highly efficient in young rodents[9,10], and our previous findings showed that Treg accelerated oligodendrocyte differentiation[27], we examined oligodendrocyte differentiation at a comparatively early time point of 7 days-post-lesion (dpl), when OPC differentiation is just starting in this model[39]. While there were no changes in overall Olig2[+] cell density, Treg expansion significantly increased the number of Olig2[+]CC1[+] differentiated oligodendrocytes in young mice (Fig. 2D–F). Next, we investigated whether Treg expansion accelerate OPC differentiation in aged mice. Taking into account that remyelination is delayed in aged mice, we first examined oligodendrocyte differentiation at 10 dpl as a relevant time point at which OPC differentiation has started in aged mice. Contrary to young mice, Treg expansion did not significantly change the density of Olig2[+] cells or Olig2[+]CC1[+] differentiated oligodendrocytes in aged mice at 10 dpl (Fig. 2G–I). We also evaluated neurofilament (NFH) and myelin basic protein (MBP) colocalization as a surrogate measure of remyelination. As for OPC differentiation, Treg expansion did not enhance axon wrapping in aged mice (Fig. 2J–L). Hence, we hypothesised that the myelin-regenerative capacity of Treg may be intrinsically impaired in aged Treg, irrespective of Treg population frequency.

### Aged Treg demonstrate an impaired intrinsic capacity to enhance myelin production

To compare the capacity of young and aged Treg to enhance OPC differentiation and myelination, we first investigated effects on oligodendrocyte development. To do this, we isolated natural Treg from the spleens and lymph nodes of young (3–4 months) and aged (14–18 months) mice (Supplementary Fig. 3A, B) and co-cultured these Treg with neonatal murine OPCs. While neither young nor aged Treg modified OPC proliferation in vitro (Fig. 3A, B), young Treg significantly increased OPC differentiation into MBP-expressing oligodendrocytes, in agreement with our previous findings of Treg secretome[27,40]. Aged Treg however, did not enhance OPC differentiation demonstrating an impairment in their capacity to support this necessary step in the process of myelin formation (Fig. 3C–E).

To determine if the inability of aged Treg to support OPC differentiation was functionally relevant to the process of myelin production, we compared the capacity of young and aged Treg to drive myelin ensheathment of axons ex vivo. To do this, we co-cultured murine organotypic brain slices with young or aged Treg for 7 days. Young Treg significantly enhanced axonal myelination compared to control slices as described before[27]. However, aged Treg failed to significantly increase myelination compared to controls (Fig. 3G). In contrast to our observations in OPC differentiation, we did not detect significant differences between young and aged Treg when evaluating myelination index (Fig. 3G) but on balance, these in vitro and ex vivo data suggest that aged Treg fail to significantly drive myelin production.

### Myelin-regenerative capacity of aged Treg is restored in a young environment

We previously reported that Treg depletion using diphtheria toxin (DT) in B6.129(Cg)-Foxp3[tm3(DTR/GFP)Ayr]/J (FoxP3-DTR)[41] mice impaired OPC differentiation and remyelination, which was rescued by the administration of wild type Treg[27]. Thus, we next compared the regenerative capacity of young and aged Treg in a model of lysolecithin-induced spinal cord myelin damage in vivo. Following endogenous GFP[+] Treg depletion (Fig. 4A, Supplementary Fig. 3C, D), we administered wild type (GFP[-]) young or aged Treg intraperitoneally and confirmed comparable detection of adoptively transferred Treg in each group (Supplementary Fig. 3E, F). We next compared the neuroregenerative capacity of adoptively transferred young and aged wild

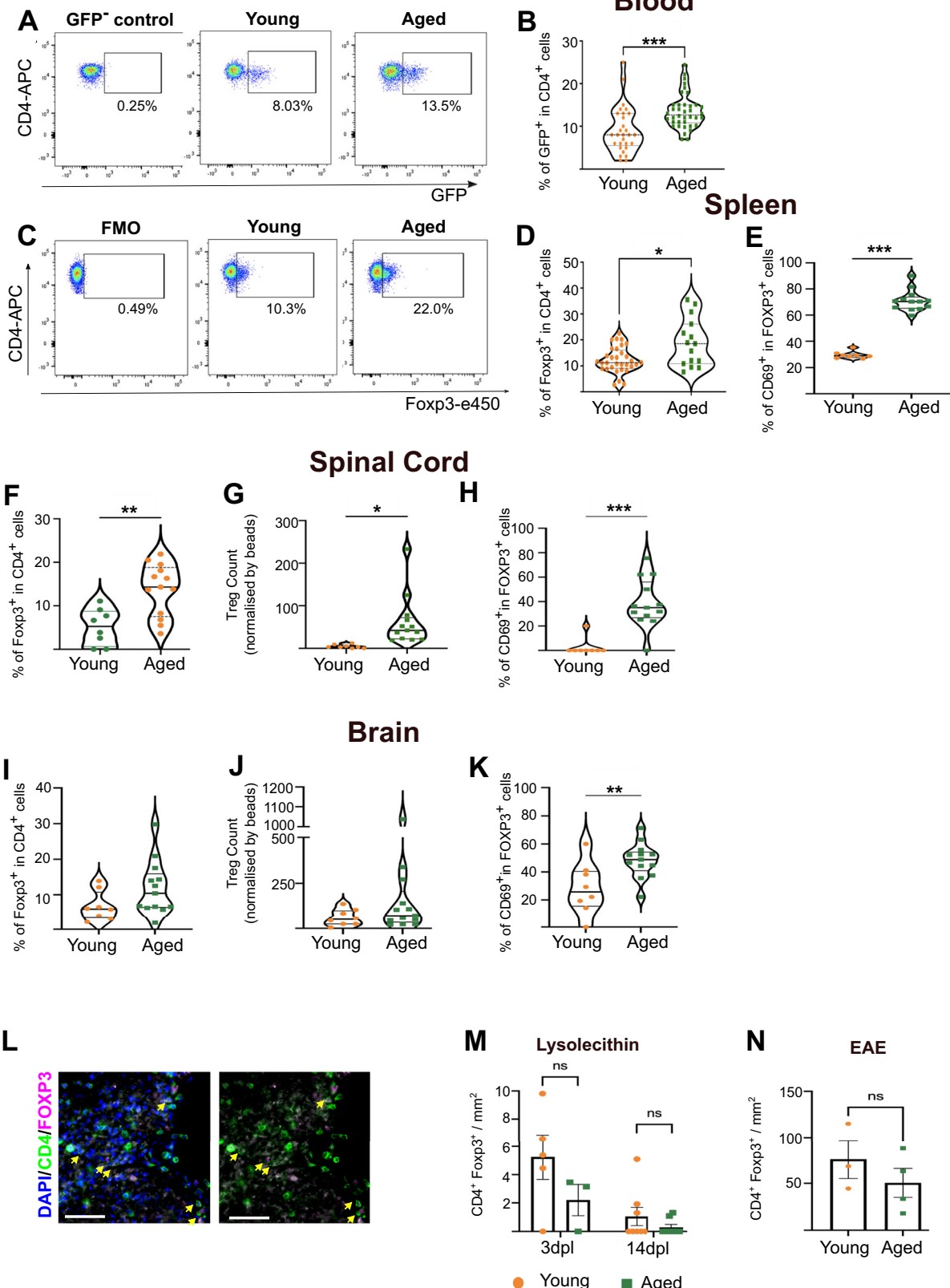

type Treg to enhance OPC differentiation during remyelination (Fig. 4A). We did not observe any differences in the burden of damage for each experimental group, indicated by the lesion area (Supplementary Fig. 3H). There was also no difference in the number of oligodendrocyte lineage cells, quantified by Olig2 immunostaining (Fig. 4B–D), or proliferating OPCs (Olig2$^+$Ki67$^+$) at 14 days post-lesion

(dpl) (Supplementary Fig. 3G, I, J). As before[27], Treg depletion impaired OPC differentiation into CC1$^+$ and ASPA$^+$ oligodendrocytes, which was rescued by adoptive transfer of wild type young Treg. Unexpectedly however, aged Treg injected into young Treg-depleted mice also rescued impaired OPC differentiation, comparably to young Treg (Fig. 4B, C, E, F). To determine if aged Treg not only rescued OPC differentiation

**Fig. 1 | Treg frequency is increased in the aged CNS. A** Flow cytometric plot and (**B**) quantification showing the proportions of natural Treg (endogenous GFP reporting Foxp3 expression in Foxp3-DTR mice) in blood of aged mice compared to young mice ($n = 29$ mice (young), $n = 38$ mice (aged), unpaired two-tailed Mann–Whitney U test ***$P$ exact < 0.0001). **C** Flow cytometric plot and (**D**) quantification of Treg proportions, identified by CD4 and Foxp3 expression, in young and aged mouse spleens ($n = 32$ mice (young), $n = 15$ mice (aged), unpaired two-tailed Mann-Whitney U test *$P = 0.019$). **E** Quantification of tissue-resident Treg proportions, identified by CD69 expression ($n = 8$ mice (young), $n = 13$ mice (aged), unpaired two-tailed Student's $t$ test after *arcsin* conversion, $t = 10.92$, ***$P < 0.0001$). **F** Violin plot-based quantification showing proportions of Treg in spinal cords of healthy young and aged mice ($n = 8$ mice (young), $n = 13$ mice (aged), unpaired two-tailed Student's $t$ test after *arcsin* conversion, $t = 3.578$, **$P = 0.002$). **G** Violin plot-based quantification showing numbers of Treg present in spinal cords of young and aged healthy mice ($n = 8$ mice (young), $n = 13$ mice (aged), unpaired two-tailed Student's $t$ test, $t = 2.716$, *$P = 0.0137$). **H** Violin plot-based quantification showing proportions of tissue-resident Treg in spinal cords of healthy young and aged mice ($n = 8$ mice (young), $n = 13$ mice (aged), unpaired two-tailed Student's $t$ test after *arcsin* conversion, $t = 4.603$, ***$P = 0.0002$). **I** Violin plot-based quantification showing proportions of Treg in brains of healthy young and aged mice ($n = 8$ mice (young), $n = 13$ mice (aged), unpaired two-tailed Student's $t$ test after *arcsin* conversion, $t = 1.725$, $P = 0.1$). **J** Violin plot-based quantification showing numbers of Treg in brains of healthy young and aged mice ($n = 8$ mice (young), $n = 13$ mice (aged), unpaired two-tailed Student's $t$ test, $t = 1.168$, $P = 0.25$). **K** Violin plot-based quantification showing proportions of tissue-resident Treg in brains of healthy young and aged mice ($n = 8$ mice (young), $n = 13$ mice (aged), unpaired two-tailed Student's $t$ test after *arcsin* conversion, $t = 2.947$, **$P = 0.0083$). **L** Immunostaining of DAPI (blue), CD4 (green) and Foxp3 (magenta) Tregs (indicated with yellow arrows) in a demyelinated lesion from a mouse with EAE (scale bar = 50 μm). **M** Quantification of the density of CD4$^+$ Foxp3$^+$ Treg in lysolecithin-induced demyelinated lesions at 3 and 14 dpl ($n = 5$ mice (young 3 dpl), $n = 3$ mice (aged 3 dpl), $n = 8$ mice (14 dpl young and aged); 2-way ANOVA, $F_{interaction} = 1.630$, $F_{dpl} = 11.49$, $F_{age} = 4.415$, Sidak's Multiple Comparison test 3dpl young vs age $P = 0.1049$, 14dpl young vs age $P = 0.7245$). **N** Quantification of the density of CD4$^+$ Foxp3$^+$ Treg in demyelinated lesions of mice with EAE at 36 days post-immunisation ($n = 3$ mice (young), $n = 4$ mice (aged), unpaired two-tailed Student's $t$ test, $t = 1.004$, $P = 0.3615$). Data are represented as mean ± SEM. Source data are provided as a Source Data File.

but also restored remyelination, we examined axonal wrapping through the quantification of circular MBP$^+$ patterns around NFH$^+$ axons as an early indication of axonal remyelination. Both, young and aged Treg administered to Treg-depleted mice, restored axonal wrapping to levels comparable to non-depleted controls (Fig. 4G–J). These findings suggest that the age-induced defect of neuroregenerative Treg functions observed in vitro/ex vivo is reversible in a young systemic environment in vivo, and therefore, potentially therapeutically targetable.

## Young Treg partially rescue OPC differentiation during remyelination in aged mice

Next, we investigated whether young Treg could rescue impaired remyelination in aged mice using Foxp3-DTR mice in which GFP⁻ young and aged Treg were adoptively transfer following DT-mediated Treg depletion. In contrast to our findings in young Treg-depleted mice, aged mice without Treg showed decreased Olig2$^+$ cell density in the demyelinating area which was not rescued neither by young nor by aged Treg (Fig. 5A, B). As per young mice, Treg depletion did not impair OPC proliferation (Fig. 5C), suggesting that the decrease in Olig2$^+$ cells likely reflects either an impaired survival or migration of oligodendrocyte lineage cells to the damaged area. The decreased density of Olig2$^+$CC1$^+$ oligodendrocytes in the absence of Treg was partially rescued by the transfer of young Treg into these animals but not by transferred aged Treg, which were significantly different to the non-depleted control (Fig. 5A, C). However, it is important to highlight that we did not detect statistically significant differences between aged Treg depleted mice receiving young or aged Treg (Fig. 5A, D) and thus, we interpret these findings with caution. A further reason to be cautious in interpreting these results is that contrary to young recipients, adoptively transferred Treg were not detectable in the classic lymphoid organs (spleen, lymph nodes) of aged Foxp3-DTR mice (Supplementary Fig. 4E, F), suggesting that the adoptively transferred cells may have a different migration pattern in aged recipients, and thus we cannot be sure that young and aged cells have the same reconstitution capacity or distribution. Additionally, since young and aged Treg failed to restore Olig2$^+$ cell density in aged mice (Fig. 5B), the age-associated remyelination impairment, even when Treg are expanded, is associated not only with impaired Treg regenerative capacity but also with a refractory aged CNS environment that may limit OPC responsiveness to Treg-derived cues, as shown for other factors previously[13].

## Ageing significantly alters Treg transcriptome

We next sought to identify how ageing impairs the myelin regenerative functions of Treg by examining gene expression differences in natural Treg purified from young and aged female mice. RNA sequencing (RNAseq) analysis showed that young and aged Treg clustered separately (Fig. 6A, C, Supplementary Data 1) as shown in the literature[42,43] and identified 1456 upregulated and 302 downregulated mRNA transcripts in aged Treg compared to young Treg (Fig. 6B, Supplementary Data 1). However, there was no difference in the expression of classic Treg markers such as *Foxp3*, *Cd4* or *Il2ra* (Supplementary Fig. 5B). Pathway analysis identified that the differentially expressed genes were associated with pathways linked to the adaptive immune response, lymphocyte-mediated immunity, phagocytosis, cell-surface receptor signalling and membrane invagination (Fig. 6D). Genes that were upregulated with ageing were enriched for the following GO biological processes: extracellular matrix organisation, cell migration, synapse assembly, axonal guidance, cell adhesion and inflammatory and immune responses (Fig. 6E). On the other hand, mRNA transcripts that were downregulated in aged Treg were associated with defence responses to virus, B cell proliferation and protein folding GO biological processes (Fig. 6E) (GO cellular component, GO Molecular function and Kyoto Encyclopedia of Genes and Genome (KEGG) database pathways analysis is shown in Supplementary Fig. 5). To identify potential mechanisms underlying how ageing may impair myelin regenerative functions of Treg, we examined expression of genes associated with the Treg tissue repair programme described recently for skin, colon, lung and adipose tisse[44]. Unexpectedly, most of the 31 Treg tissue repair programme hallmark/signature genes[44] were in fact upregulated with ageing, except for *Klrb1, Ccr5, Selplg, Bach2, Rad50, Tfrc, Cd84* and *Cd164* (Fig. 6F, G). No difference in expression was detected in these hallmark genes between male and female natural Treg, showing that sex does not alter the expression of genes associated with the tissue repair programme (Supplementary Fig. 5C, D). These data suggest that the mechanism(s) involved in Treg-driven myelin regeneration capacity that are impaired with ageing, are likely different to mechanisms from the Treg tissue repair programme described by Delacher et al.[44].

## Mcam and Itga2 are associated with Treg capacity to drive oligodendrocyte formation

To narrow down age-induced changes in Treg that diminish their capacity to induce OPC differentiation, we first investigated whether the putative mechanisms are soluble or contact-mediated; the latter being suggested by the close physical association of Treg with OPCs observed in co-cultures (Fig. 7A) and in EAE-induced, demyelinated lesions in vivo (Fig. 7B). Young natural Treg were co-cultured either directly in contact with OPC, or in the same well but separated using a

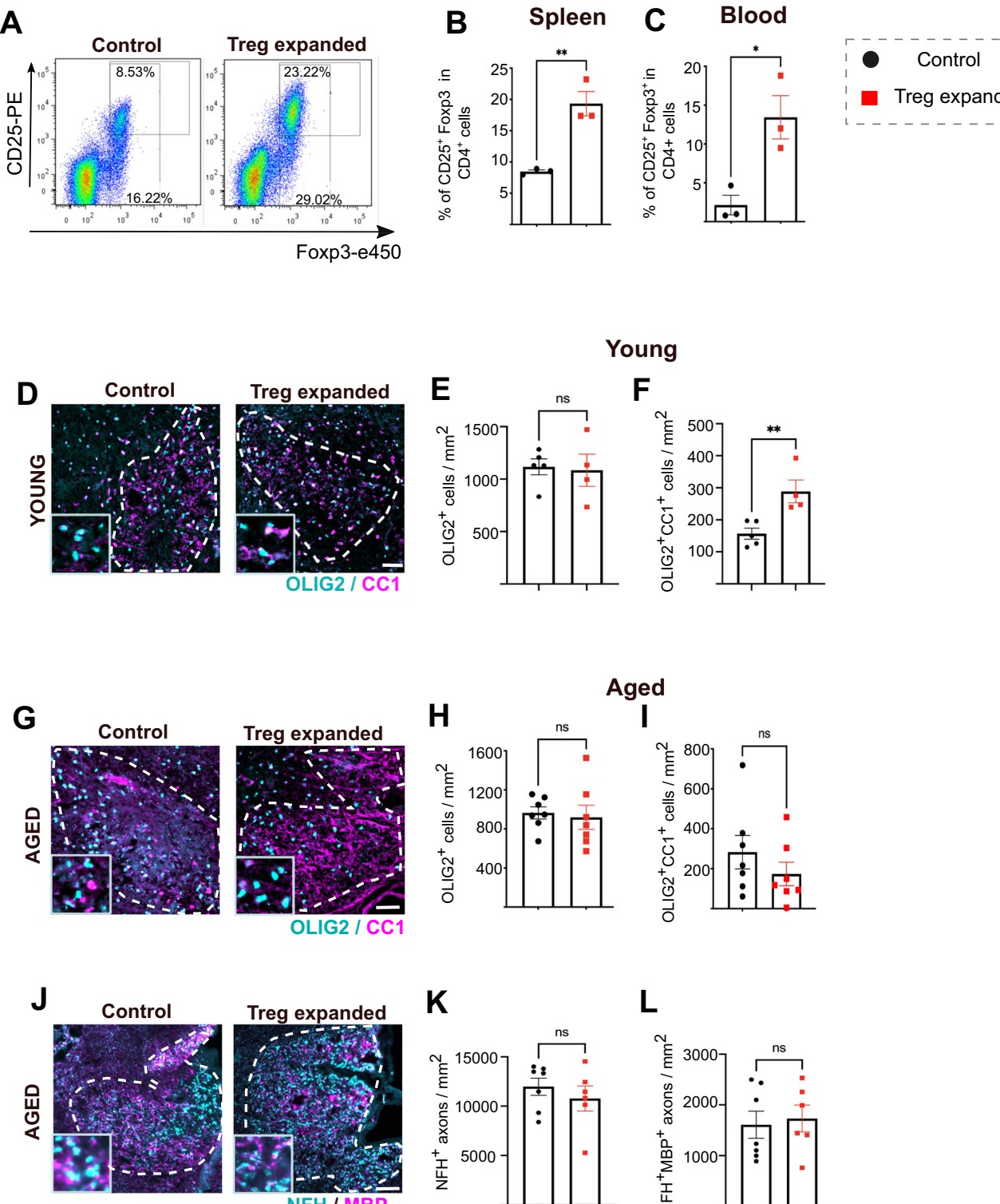

transwell, which allowed for the exchange of secreted molecules but not direct OPC-Treg contact. While young Treg co-cultured directly with OPCs enhanced OPC differentiation into MBP⁺ oligodendrocytes, this pro-differentiating effect was lost when Treg were physically separated from OPCs (Fig. 7C, D). To identify potential cell-to-cell contact mechanisms responsible for Treg-driven OPC differentiation that may be lost with ageing, we combined our aged Treg RNA sequencing data with a publicly available RNAseq dataset that compares between OPCs, newly differentiated and differentiated oligodendrocytes[45]. We hypothesised that ligand-receptor interactions

between Treg and plasma membrane receptors enriched in OPCs underlie the pro-differentiation effect of Treg on OPC. We prioritised ligand-receptor partnerships between Treg and OPCs that were impacted by ageing, to identify mechanisms that could explain the lost myelin regenerative capacity of aged Treg. We then performed a protein-protein interaction analysis using String[46] (Fig. 7F) and prioritised plasma membrane proteins that had the highest decrease in expression for aged Treg and the highest number of potential binding partners amongst OPC-enriched receptors compared to oligodendrocytes (Fig. 7G). We identified the following key candidates: *Ccr7*,

**Fig. 2 | Expansion of Treg does not rescue impaired OPC differentiation in aged mice. A** Flow cytometric plot showing CD25 and Foxp3 expression of control aged mice and aged mice treated with intraperitoneal injection of IL-2/anti-IL-2 complexes. **B** Quantification of CD25⁺Foxp3⁺ natural Treg proportion in a CD4⁺ T cell population from the spleen control aged mice and aged mice treated with intraperitoneal injection of IL-2/anti-IL-2 complexes ($n = 3$, unpaired two-tailed Student's *t* test after *arcsin* conversion, $t = 5.49$, **$P = 0.0054$). **C** Quantification of CD25⁺Foxp3⁺ natural Treg proportion in a CD4⁺ T cell population from the blood control aged mice and aged mice treated with intraperitoneal injection of IL-2/anti-IL-2 complexes ($n = 3$, unpaired two-tailed Student's *t* test after *arcsin* conversion, $t = 3.679$, *$P = 0.0212$). **D** Representative images of OLIG2 (cyan) and CC1 (magenta) immunostaining in young control and Treg expanded demyelinated lesions at 7dpl (scale bar = 50 μm, lesion area demarcated with the dotted line). **E** Quantification of OLIG2⁺ cell density in the demyelinated area of young control and Treg expanded mice ($n = 5$ mice (control), $n = 4$ mice (Treg expanded), unpaired two-tailed Student's *t* test, $t = 0.2018$, $P = 0.8458$). **F** Quantification of OLIG2⁺CC1⁺ cell density in the demyelinated area of young control and Treg expanded mice ($n = 5$ mice (young), $n = 4$ mice (Treg expanded), unpaired two-tailed Student's *t* test, $t = 3.59$,

**$P = 0.0089$). **G** Representative images of OLIG2 (cyan) and CC1 (magenta) immunostaining in aged control and Treg expanded demyelinated lesions at 10dpl (scale bar = 50 μm, lesion area demarcated with the dotted line). **H** Quantification of OLIG2⁺ cell density in the demyelinated area of aged control and Treg expanded mice ($n = 7$ mice (control and Treg expanded), unpaired two-tailed Student's *t* test, $t = 0.3267$, $P = 0.7495$). **I** Quantification of OLIG2⁺CC1⁺ cell density in the demyelinated area of aged control and Treg expanded mice ($n = 7$ mice (control and Treg expanded), unpaired two-tailed Student's *t* test, $t = 1.07$, $P = 0.3055$).
**J** Representative images of NFH (cyan) and MBP (magenta) immunostaining in aged control and Treg expanded demyelinated lesions at 10dpl (scale bar = 50 μm, lesion area demarcated with the dotted line). **K** Quantification of NFH⁺ axon density in the demyelinated area of aged control and Treg expanded mice ($n = 7$ mice (young), $n = 6$ (aged), unpaired two-tailed Student's *t* test, $t = 0.8018$, $P = 0.4397$).
**L** Quantification of NFH⁺MBP⁺ axon density in the demyelinated area of aged control and Treg expanded mice ($n = 7$ mice (young), $n = 6$ (aged), unpaired two-tailed Student's *t* test, $t = 0.3268$, $P = 0.75$). Data are represented as mean ± SEM. Source data are provided as a Source Data File.

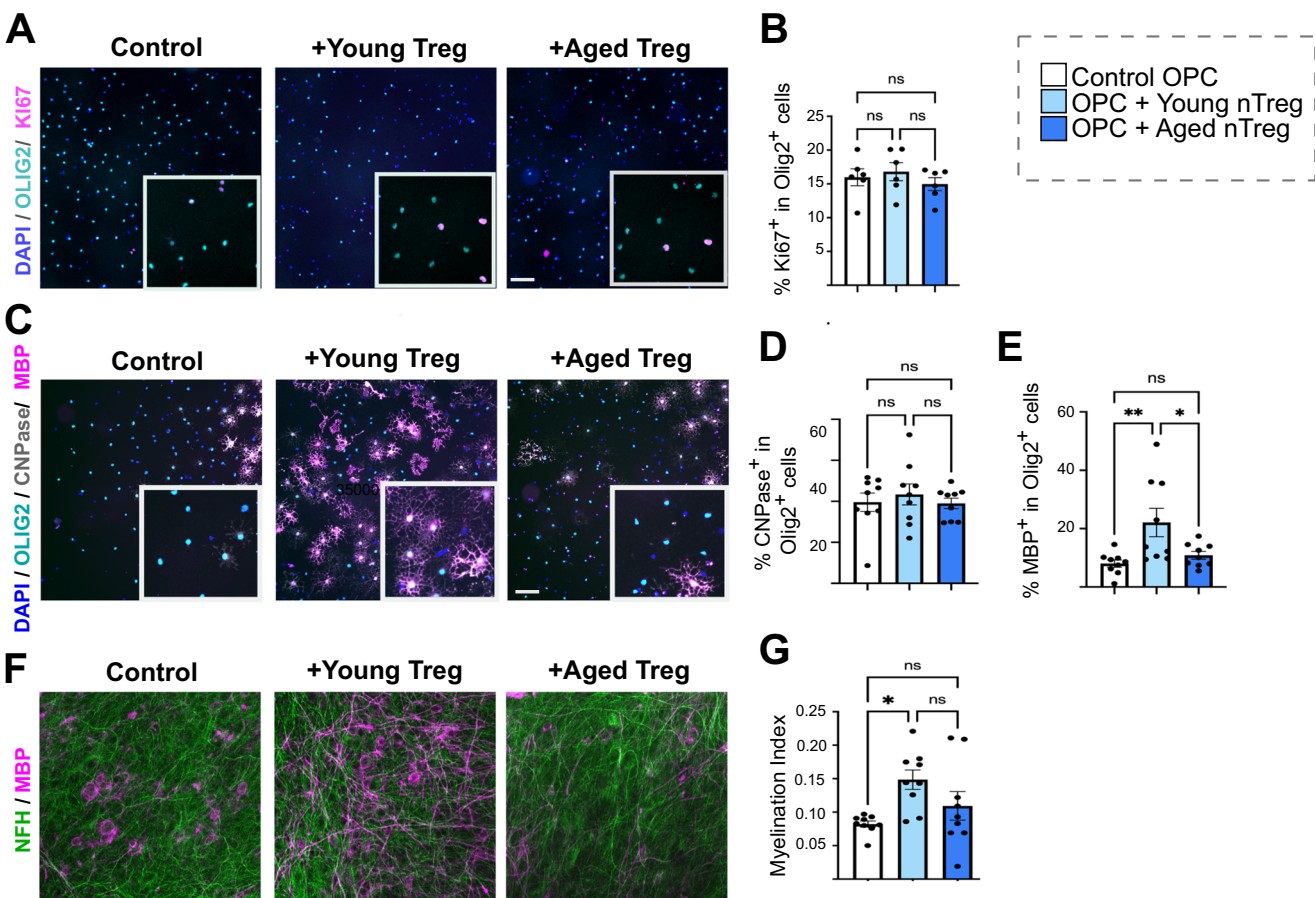

**Fig. 3 | Aged natural Treg demonstrate impaired capacity to drive OPC differentiation and myelination in vitro. A** Representative images of OPCs co-cultured with young and aged natural Treg and immunostained for OLIG2 (cyan) as a pan oligodendrocyte lineage marker and the proliferation marker Ki67 (magenta) (scale bar = 100 μm). **B** Quantification of OPC proliferation when co-cultured with young and aged natural Treg ($n = 6$ mice, 2 independent experiments, 1-way ANOVA after *arcsin* conversion, $F = 0.4279$, $P_{control vs young} = 0.9487$, $P_{control vs aged} = 0.9121$, $P_{young vs aged} = 0.6418$). **C** Representative images of OPCs co-cultured with young and aged natural Treg and immunostained for OLIG2 (cyan) and differentiation markers CNP (grey) and MBP (magenta) (scale bar = 100 μm). **D** Quantification of the proportion of OPCs reaching early stage-differentiation when exposed to young and aged natural Treg, as indicated by CNPase staining ($n = 9$ mice, 3 independent experiments, 1-way ANOVA after *arcsin* conversion, $F = 0.8282$,

$P_{control vs young} = 0.8925$, $P_{control vs aged} = 0.9995$, $P_{young vs aged} = 0.8439$).
**E** Quantification of the proportion of OPCs expressing late-stage differentiation marker MBP in control OPCs and OPCs treated with young and aged natural Treg ($n = 9$ mice, 3 independent experiments, 1-way ANOVA, Sidak's multiple comparisons test after *arcsin* conversion, $F = 3.87$, **$P_{control vs young} = 0.0094$, $P_{control vs aged} = 0.8951$, *$P_{young vs aged} = 0.0422$). **F** Immunohistochemistry of control and young and aged Treg treated cerebellar slices (MBP, magenta and NFH, green, scale bar = 100 μm). **G** Quantification of myelination index (ratio between MBP and NFH colocalization area and NFH area) in neonatal cerebellar slices ($n = 9$ mice, 1 brain slice per mouse, 3 independent experiments, 1-way ANOVA after *arcsin* conversion, Sidak's multiple comparison tests, $F = 4.946$, *$P_{control vs young} = 0.0147$, $P_{control vs aged} = 0.5215$, $P_{young vs aged} = 0.2203$). Data are represented as mean ± SEM. Source data are provided as a Source Data File.

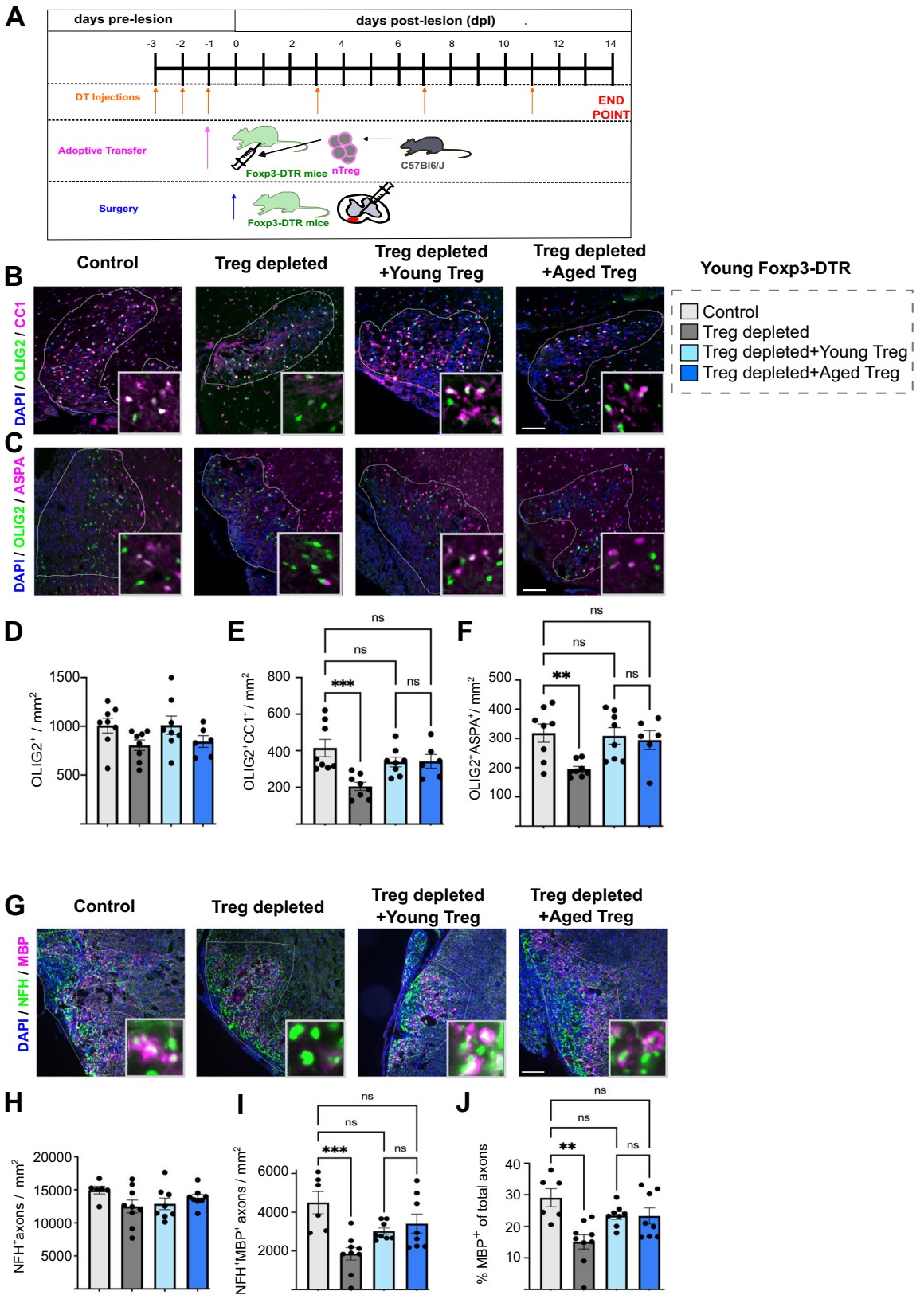

*Itga2* (CD49b), *Klrb1c* (NK1.1), *Ly6c1, Mcam* (CD146) and *Sell* (CD62L) (Fig. 5F–H). Next, we used a loss-of-function approach to investigate whether these candidates may confer the lost regenerative capacity of aged Treg, namely, CD62L, NK1.1, LY6C, MCAM (CD146) and ITGA2 (CD49b). Of these five targets, antibody-mediated blocking of MCAM (CD146) and ITGA2 (CD49b) on Treg prior to Treg-OPC co-culture

inhibited the capacity of young Treg to drive OPC differentiation in vitro as indicated by the fold change in the percentage of MBP⁺ cells and the MBP⁺ area per well (Fig. 8A–C, Supplementary Fig. 6A, B). This suggests that MCAM and ITGA2 are previously unidentified mediators of pro-remyelinating functions of Treg and are downregulated with age.

**Fig. 4 | Myelin-regenerative capacity of aged Treg is restored in a young environment. A** Diagram explaining the experimental design of in vivo Treg depletion, Treg adoptive transfer and spinal cord demyelination. Representative images of immunostaining identifying oligodendrocytes by the co-localisation of the pan-oligodendrocyte lineage marker OLIG2 (green) with CC1 (magenta, **B**) or ASPA (magenta, **C**) at 14 dpl (scale bar = 100 μm, demyelination area is indicated by the white line). **D** Bar graph showing the quantification of total number of oligo-dendrocyte lineage cells in the demyelinated lesions of PBS control mice ($n = 8$ mice), natural Treg-depleted mice ($n = 8$ mice) and mice depleted of endogenous Treg that received young ($n = 8$ mice) or aged natural Treg ($n = 6$ mice) by adoptive transfer (1-way ANOVA, Sidak's multiple comparisons test, $F = 2.134$, $P_{PBS\ vs\ depleted} = 0.2143$, $P_{PBS\ vs\ young} > 0.9999$, $P_{PBS\ vs\ aged} = 0.4726$, $P_{young\ vs\ aged} = 0.4562$). **E** Bar graph showing the quantification of total number of CC1-expressing oligoden-drocytes in the demyelinated lesions of PBS control mice ($n = 8$ mice), natural Treg-depleted mice ($n = 8$ mice) and mice depleted of endogenous Treg that received young ($n = 8$ mice) or aged natural Treg ($n = 6$ mice) by adoptive transfer (1-way ANOVA, Sidak's multiple comparisons test, $F = 6.773$, $***P_{PBS\ vs\ depleted} = 0.0006$, $P_{PBS\ vs\ young} = 0.4065$, $P_{PBS\ vs\ aged} = 0.5240$, $P_{young\ vs\ aged} > 0.9999$). **F** Bar graph showing the quantification of total number ASPA-expressing oligodendrocytes in the demyelinated lesions of PBS control mice ($n = 8$ mice), natural Treg-depleted mice ($n = 8$ mice) and mice depleted of endogenous Treg that received young ($n = 8$ mice) or aged natural Treg ($n = 6$ mice) by adoptive transfer (1-way ANOVA, Sidak's multiple comparisons test, $F = 4.901$, $**P_{PBS\ vs\ depleted} = 0.0088$, $P_{PBS\ vs\ young} =$

$0.9985$, $P_{PBS\ vs\ aged} = 0.9597$, $P_{young\ vs\ aged} = 0.9933$). **G** Representative images of immunostaining for neurofilament-H (NFH, green) and MBP (magenta) to quantify myelin wrapping as an early marker of remyelination at 14 dpl (scale bar = 100 μm, demyelination area is indicated by the white line). **H** Quantification shows the total number of axons in the demyelinated lesions of PBS control mice ($n = 6$ mice), natural Treg-depleted mice ($n = 9$ mice) and mice depleted of endogenous Treg that received young ($n = 8$ mice) or aged natural Treg ($n = 8$ mice) by adoptive transfer (1-way ANOVA, Sidak's multiple comparisons test, $F = 1.689$, $P_{PBS\ vs\ depleted} = 0.1740$, $P_{PBS\ vs\ young} = 0.3410$, $P_{PBS\ vs\ aged} = 0.8304$, $P_{young\ vs\ aged} = 0.8793$). **I** Quantification shows the density of MBP-wrapped axons in the demyelinated lesions of PBS control mice ($n = 6$ mice), natural Treg-depleted mice ($n = 9$ mice) and mice depleted of endogenous Treg that received young ($n = 8$ mice) or aged natural Treg ($n = 8$ mice) by adoptive transfer (1-way ANOVA, Sidak's multiple comparisons test, $F = 7.342$, $***P_{PBS\ vs\ depleted} = 0.0004$, $P_{PBS\ vs\ young} = 0.0738$, $P_{PBS\ vs\ aged} = 0.2766$, $P_{young\ vs\ aged} = 0.9273$). **J** Quantification shows the percentage of MBP-wrapped axons from the total number of axons in the demyelinated lesions of PBS control mice ($n = 6$ mice), natural Treg-depleted mice ($n = 9$ mice) and mice depleted of endogenous Treg that received young ($n = 8$ mice) or aged natural Treg ($n = 8$ mice) by adoptive transfer (Kruskal-Wallis test, Dunn's multiple comparisons test, Kruskal Wallis statistic = 14.16, $**P_{PBS\ vs\ depleted} = 0.0015$, $P_{PBS\ vs\ young} > 0.9999$, $P_{PBS\ vs\ aged} = 0.3523$, $P_{young\ vs\ aged} > 0.9999$). Data are represented as mean ± SEM. Source data are provided as a Source Data File.

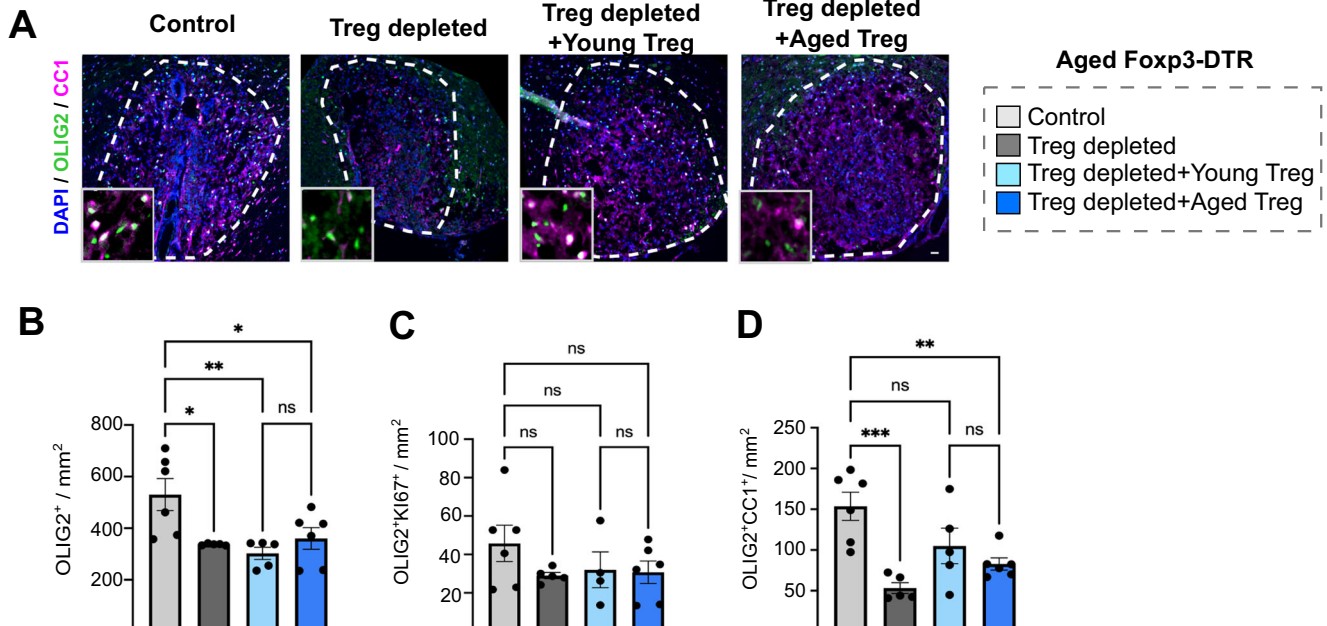

**Fig. 5 | Young Treg only partially rescue oligodendrocyte differentiation in aged Treg-depleted mice. A** Representative images of immunostaining identifying oligodendrocytes by the co-localisation of the pan-oligodendrocyte lineage marker OLIG2 (green) with CC1 (magenta) at 14 dpl (scale bar = 100 μm, demyelination area is indicated by the white dotted line). **B** Bar graph shows the quantification of total number of oligodendrocyte lineage cells in the demyelinated lesions of PBS-treated control aged mice ($n = 6$ mice), natural Treg-depleted aged mice ($n = 5$ mice) and aged mice depleted of endogenous Treg that received young ($n = 5$ mice) or aged natural Treg ($n = 6$ mice) by adoptive transfer (1-way ANOVA, Sidak's multiple comparisons test, $F = 5.859$, $*P_{PBS\ vs\ depleted} = 0.0199$, $**P_{PBS\ vs\ young} = 0.0057$, $*P_{PBS\ vs\ aged} = 0.0343$, $P_{young\ vs\ aged} = 0.8230$). **C** Bar graph shows the quantification of total number of Ki67+ proliferating OPCs in the demyelinated lesions of PBS-

treated control aged mice ($n = 6$ mice), natural Treg-depleted aged mice ($n = 5$ mice) and aged mice depleted of endogenous Treg that received young ($n = 4$ mice) or aged natural Treg ($n = 6$ mice) by adoptive transfer (1-way ANOVA, Sidak's multiple comparisons test, 1.207, $P_{PBS\ vs\ depleted} = 0.3939$, $P_{PBS\ vs\ young} = 0.6326$, $P_{PBS\ vs\ aged} = 0.4502$, $P_{young\ vs\ aged} > 0.9999$). **D** Bar graph shows the quantification of CC1+ oligodendrocytes in the demyelinated lesions of PBS-treated control aged mice ($n = 6$ mice), natural Treg-depleted aged mice ($n = 5$ mice) and aged mice depleted of endogenous Treg that received young ($n = 5$ mice) or aged natural Treg ($n = 6$ mice) by adoptive transfer (1-way ANOVA, Sidak's multiple comparisons test, $F = 8.562$, $***P_{PBS\ vs\ depleted} = 0.0005$, $P_{PBS\ vs\ young} = 0.1168$, $**P_{PBS\ vs\ aged} = 0.0084$, $P_{young\ vs\ aged} = 0.7576$). Data are represented as mean ± SEM. Source data are pro-vided as a Source Data File.

 Next, we sought to determine whether MCAM and ITGA2 would directly promote OPC differentiation by treating neonatal OPCs with recombinant ITGA2 and MCAM. While treatment with rMCAM for up to 6 days in vitro did not affect OPC proliferation or differentiation (Supplementary Fig. 6C, E–G), rITGA2 directly enhanced OPC

differentiation by day 3 (Fig. 8D–G). rITGA2 did not significantly change the proportion of proliferating OPC (Supplementary Fig. 6D) or the proportion of MBP-expressing oligodendrocytes (Fig. 8D, E). However, we observed an increased complexity of oligodendrocyte morphology in rITGA2-treated cultures as indicated by the enhanced

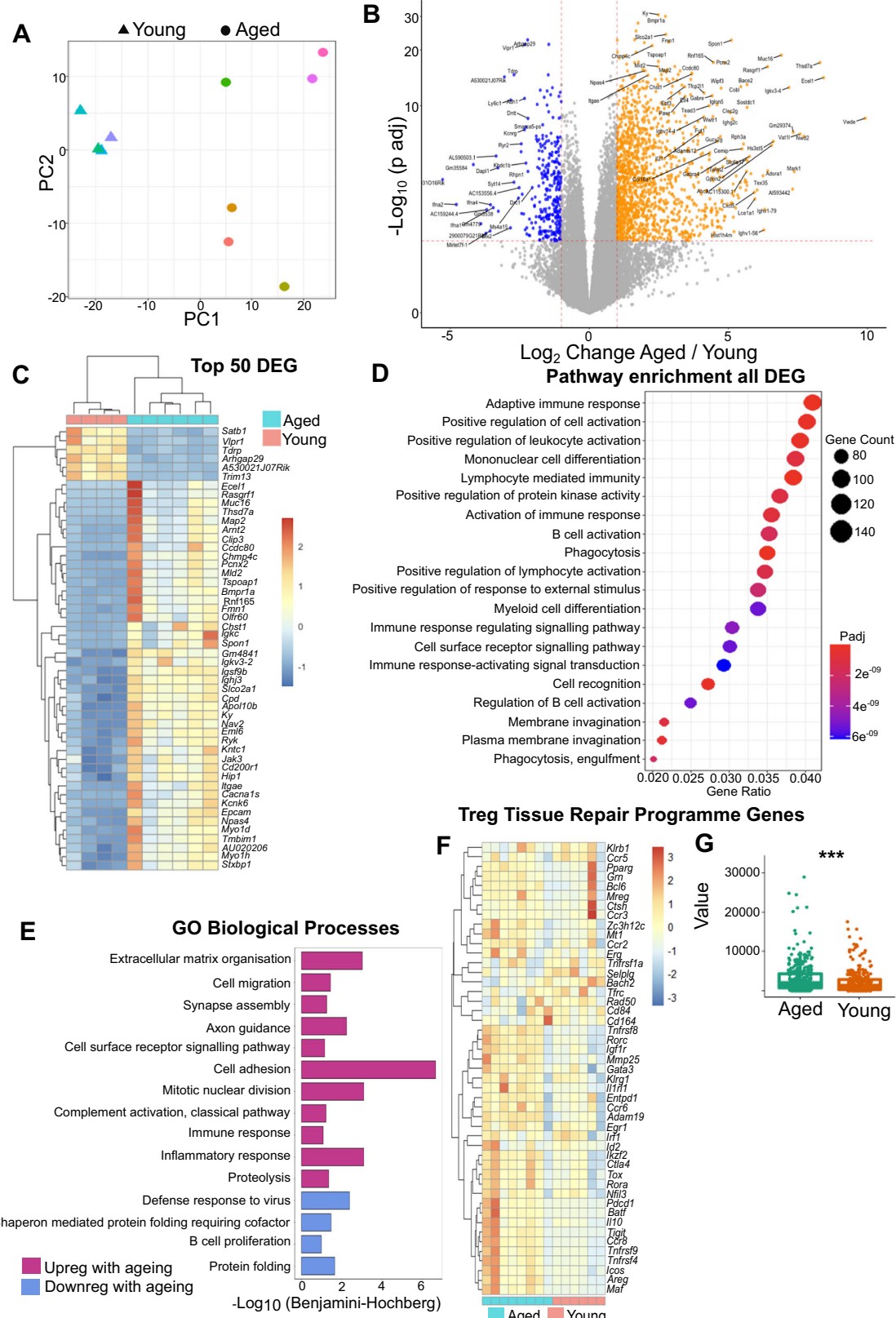

number of intersections detected by Sholl analysis on ITGA2-treated OPCs (Fig. 8F, G). This suggests that oligodendrocytes in rITGA2 treated cultures are at later stage of development which aligns with an acceleration of differentiation. Therefore, while rITGA2 directly promoted OPC differentiation, rMCAM did not, suggesting that the function of this latter cell adhesion molecule in Treg-mediated OPC

differentiation may be to facilitate OPC-Treg interaction and potentially to activate other pro-differentiative mechanisms rather than itself directly driving OPC differentiation.

To determine whether a young environment could boost MCAM and ITGA2 expression by aged Treg, we adoptively transferred young and aged Treg expressing red fluorescent protein (RFP+) obtained

**Fig. 6 | Ageing significantly alters natural Treg transcriptome. A** Principal component analysis demonstrating the clustering differences between young and aged natural Treg. **B** Volcano plot demonstrating 1456 genes upregulated and 302 genes downregulated genes in aged natural Treg relative to young Treg ($n = 4$ mice young, $n = 6$ mice aged, Wald test, Bonferroni multiple test correction). **C** Heatmap demonstrating hierarchical clustering of the top 50 differentially expressed genes between young and aged Treg. **D** Graph showing the pathways enriched amongst differentially expressed genes. **E** Bar graph highlighting the GO biological processes associated with genes that are upregulated or downregulated in aged Treg.

**F** Heatmap showing the normalised count values for the Treg tissue repair programme identified by Delacher et al.[44]. **G** Box plot showing the Treg tissue repair programme signature genes score in aged and young natural Treg ($n = 4$ mice young, $n = 6$ mice aged, 47 genes associated to tissue repair Treg signature considered, two-tailed, unpaired Wilcoxon test, \*\*\*$P = 0.00021$; Data shown as box plots in which the centre line denotes the median value (50th percentile), while the box contains the 25th to 75th percentile datasets). Source data are provided as a Source Data File.

from Foxp3-RFP mice (C57BL/6-*Foxp3*[tm1Flv]/J Foxp3-IRES-mRFP) into young, Treg-depleted mice and assessed ITGA2 and MCAM expression in spleen and lymph node after 11 days by flow cytometry (Supplementary Fig. 7A, B). Due to technical challenges and a lack of further aged FoxP3 reporter mice, only $n = 2–3$ was possible in this study which limits interpretation of statistical significance of our findings. The proportions of MCAM+ or ITGA2+ cells within total Foxp3-RFP+ populations were low in basal conditions and were significantly increased in both aged and young Treg upon adoptive transfer (Supplementary Fig. 7C, F). To examine the extent of expression of each marker on a per cell basis, we quantified median fluorescent intensity of MCAM+RFP+ and ITGA2+RFP+ Treg and observed decreased expression of both markers in aged Treg compared to young Treg, in agreement with our RNAseq findings. Following adoptive transfer into young mice, expression of ITGA2 was fully restored to levels comparable to that of young Treg (Supplementary Fig. 7C–E), while MCAM in aged Treg was partially restored (Supplementary Fig. 7F–H). These findings suggest that ITGA2 and MCAM are Treg-expressed proteins that directly or indirectly support OPC differentiation, are decreased with age and expression of which can be restored when aged Treg are adoptively transferred into a young environment.

## Discussion

Ageing is one of the strongest risk factors associated with the transition from relapsing-remitting to secondary progressive MS[12], a phase that is characterised by the accumulation of neurodegeneration and irreversible disability. This is partly due to impaired remyelination with age, which was first described in animal models of remyelination[9,10] but has now also been observed in MS patients[8,11]. Several factors have been linked to age-associated myelin regeneration failure, such as impaired myelin debris phagocytosis by microglia and macrophages[19,20,47], reduced intrinsic capacity of OPCs to differentiate into myelin-forming oligodendrocytes[13,17] and a refractory CNS environment with increased stiffness and changes in extracellular matrix[16,17]. Therapies that enhance remyelination and prevent age-related neurological decline are an urgent unmet need. Beyond their classic immune-modulatory and anti-inflammatory role, Treg have recently emerged as key regulators of tissue homeostasis and regeneration in several tissues[27], including the CNS, where we have previously identified Treg as key promoters of remyelination[27]. However, how ageing affects the capacity of Treg to enhance myelin regeneration had not been addressed.

Peripheral Treg[30,42,48], as well as Treg and tissue-resident Treg increase with age, a somewhat paradoxical finding given the tissue-regenerative functions of Treg and the impairment of tissue regeneration with age. Here, we further expanded the murine Treg population in vivo and showed that despite this increased cell number, aged-associated remyelination failure was not rescued. These data suggest that with ageing, Treg have an intrinsically impaired myelin-regenerative capacity. Ageing is known to enhance Treg senescence, limit Treg proliferation[30,42] and there are conflicting reports on the effect of ageing on Treg immunosuppressive capacity[30,42]. Previous work on muscle regeneration showed that aged Treg have an impaired migratory capacity that hinders muscle regeneration[36]. Additionally, in a model of influenza infection, aged Treg demonstrated an impaired

capacity to support lung repair, which was associated with a cell autonomous impairment in their regenerative programme[43]. Here, we sought to determine whether aged Treg have an intrinsically impaired capacity to drive CNS myelin regeneration combining in vitro, ex vivo and in vivo approaches.

Purified aged Treg failed to significantly drive OPC differentiation into MBP-expressing oligodendrocytes or the myelination of organotypic cerebellar slices, which was in striking contrast to young Treg, which robustly drove both biological processes. These data suggest that ageing is associated with a cell-intrinsic impairment of the regenerative capacity of Treg in CNS myelin regeneration. Given that IL-2-mediated Treg expansion in vivo is in clinical trial for MS (clinical trial number NCT02424396)[49], our findings are of translational importance as they suggest that Treg expansion as a potential pro-remyelination therapy may be limited in older populations.

We next investigated whether the regenerative capacity of aged Treg could be restored, and therefore therapeutically targetable. Surprisingly, aged Treg rescued OPC differentiation and myelin wrapping similarly to that of young Treg in an in vivo model of young spinal cord remyelination. In contrast to findings in the lung[43], the CNS remyelinating capacity of Treg can be partially restored by a young environment. Interestingly, in lesions of aged mice lacking Treg, young Treg partially restored Olig2+CC1+ oligodendrocyte density, while aged Treg did not, suggesting that indeed young Treg have a more potent regenerative capacity than aged Treg. However, neither young nor aged Treg restored Olig2+ density or remyelination caused by depletion of endogenous Treg, suggesting that with age, the failure of Treg to drive remyelination is a combination of both an intrinsically impaired regenerative capacity of Treg and an aged CNS that is refractory to Treg-derived pro-differentiation signals. Previous studies identified a tissue repair transcriptomic signature in the skin, adipose tissue, and the lung[44]. Unexpectedly, very few of these hallmark Treg tissue repair genes were downregulated in aged Treg in our transcriptomic profiling studies. These data indicate that although some Treg regenerative mechanisms may be common across tissues, other mechanisms may be tissue-specific or process- specific[50].

We recognised that the impairment of myelin regenerative Treg functions with age presented an avenue to discover mechanisms of how Tregs support CNS remyelination and the potential to identify targets that may be of therapeutic relevance to boost remyelination. Indeed, using a combined approach of Treg/OPC transcriptomic dataset cross-referencing with in vitro antibody blocking assays, we identified MCAM and ITGA2 as candidate mediators of Treg-driven OPC differentiation. These two candidates were downregulated in aged compared to young Treg, likely interact directly with OPCs and inhibit young Treg-driven OPC differentiation when blocked in vitro. Transfer of aged Treg into young mice boosted expression of ITGA2 and to a lesser extent of MCAM by Treg in preliminary studies, suggesting that these are modifiable factors that can be restored in aged Treg. Furthermore, recombinant ITGA2, but not MCAM, promoted OPC differentiation in vitro suggesting that these proteins have direct and indirect roles in Treg-driven OPC differentiation.

Our results identify ITGA2 as a mediator of Treg-driven OPC differentiation, that is decreased with age, and can directly promote OPC differentiation. This aligns with previous reports showing that other

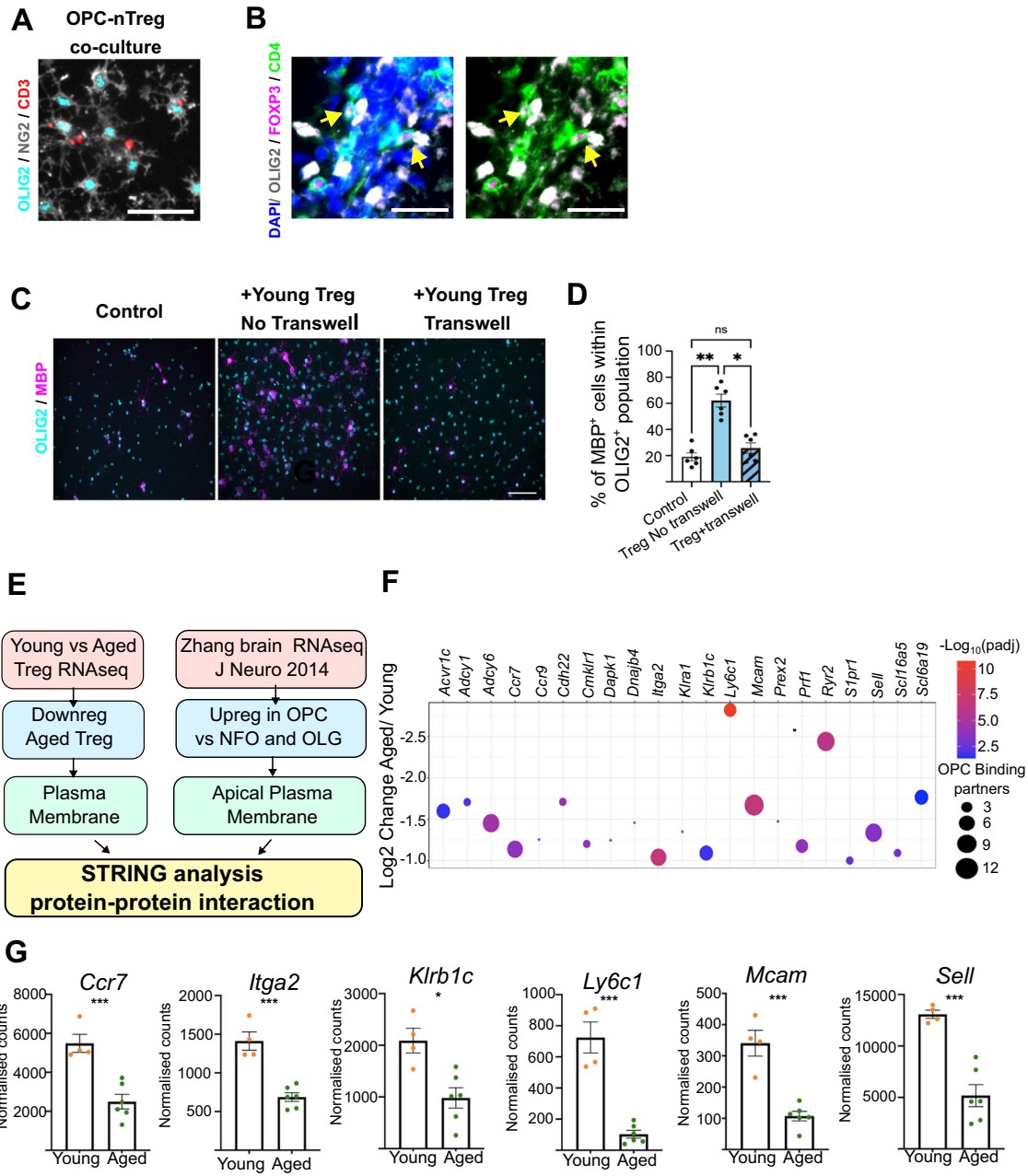

**Fig. 7 | Natural Treg-driven OPC differentiation in vitro is contact-dependent.**
**A** Representative image of immunostaining showing cell-to-cell contact between OPCs and Treg in OPC-Treg co-cultures in vitro. OPCs are identified by the co-staining of OLIG2 (cyan) and NG2 (grey), while Treg are identified by CD3 (red) (scale bar = 50 µm). **B** Representative images showing Treg (CD4⁺ (green), Foxp3⁺ (magenta)) in close proximity or direct contact with OLIG2⁺ oligodendrocyte lineage cells in demyelinated lesions in mice with EAE, highlighted by the yellow arrows (scale bar = 25 µm). **C** Representative images of immunostaining of OPCs directly co-cultured with young Treg and OPCs cultured with young Treg in a transwell (OLIG2 (cyan) and MBP (magenta), scale bar = 100 µm). **D** Quantification of MBP-expressing oligodendrocytes in control OPCs, OPCs directly co-cultured with young Treg and OPCs cultured with young Treg in a transwell ($n = 6$ mice, 2 independent experiments, Kruskal-Wallis test, Dunn's multiple comparison's test, KW

statistic = 12.23, **$P_{control\ vs\ Treg\ no\ transwell}$ = 0.0022, $P_{control\ vs\ Treg\ transwell}$ > 0.9999, *$P_{Treg\ no\ transwell\ vs\ Treg\ transwell}$ = 0.0415). **E** Diagram summarising bioinformatic approaches to identify protein–protein interactions between OPCs and Treg. **F** Graph showing 21 protein candidates expressed in the Treg plasma membrane, that are downregulated in aged Treg and have potential bindings partners enriched in OPCs vs oligodendrocytes ($n = 4$ mice young, $n = 6$ mice aged, Wald test, Bonferroni multiple test correction). Log2 Change, -Log10 (Padj) and the number of OPC binding partners are indicated (see legend). **G** Bar graphs showing RNAseq normalised count values for the top 6 candidates ($n = 4$ mice young, $n = 6$ mice aged, Wald test, Bonferroni multiple test correction, ***$P_{Ccr7}$ = 0.0002, ***$P_{Itga2}$ < 0.00001, *$P_{Klrb1c}$ = 0.0215, ***$P_{Ly6c1}$ < 0.00001, ***$P_{Mcam}$ < 0.00001, ***$P_{Sell}$ = 0.0004). Data are represented as mean ± SEM. Source data are provided as a Source Data File.

integrins are involved in oligodendrocyte development[51] and shows that Treg are additional cellular contributors to integrin-mediated OPC differentiation. MCAM has previously been described as a cell-adhesion molecule contributing to brain inflammation by facilitating pathogenic T cell extravasation into the CNS[52–54] and here, we identify a

putative regenerative function of MCAM expressed by Treg. Together these data contribute to the growing evidence that Treg express MCAM[55,56] and suggests a potential regenerative role mediating Treg-stem cell interactions. Even though our data show that, in vitro, non-activated natural Treg require direct OPC-Treg contact to drive OPC

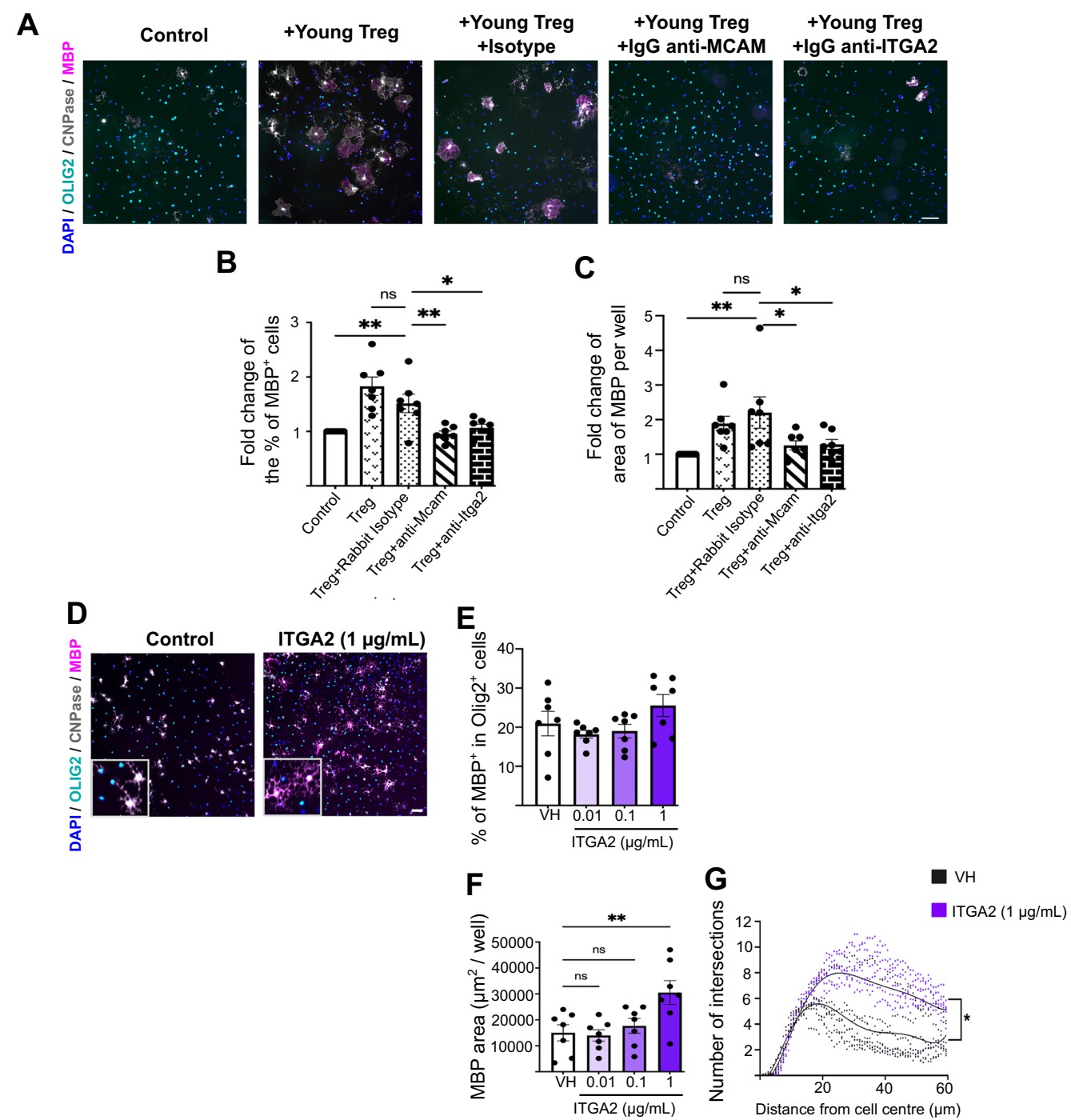

differentiation, previous work from our laboratory showed that activated Treg secretome also enhanced OPC differentiation[27]. Thus, Treg-mediated regenerative capacity likely involves a range of signalling molecules that regulate OPC-Treg interaction as well as subsequent OPC differentiation and remyelination, providing a range of potential mechanisms to explore as therapeutic targets for remyelination therapy in the future.

Collectively, these studies have identified that ageing is associated with a cell-intrinsic impairment of the myelin-regenerative functions of Treg. This discovery increases the range of remyelination-associated mechanisms that are impaired with age[9,10,13,17–20,47]. Encouragingly, we have shown that this impairment can be rescued in a young environment in vivo. We have also identified two mediators of Treg-driven OPC differentiation, MCAM and ITGA2. This study has identified prospective mechanisms involved in CNS regeneration which need to be

further investigated to determine if they hold therapeutic potential for patients with demyelinating diseases.

## Methods

### Animals

All mice were on a C57BL/6J background and were either bred in-house or purchased from Charles River Laboratories, UK. Foxp3$^{tm3(DTR/GFP)Ayr}$/J (Foxp3-DTR) mice were a kindly provided by Prof. Alexander Rudensky (Memorial Sloan Kettering Institute, New York) and C57BL/6-Foxp3$^{tm1Flv}$/J Foxp3-IRES-mRFP (Foxp3-RFP) mice were a kind gift from Dr. Rebecca Ingram. Neonatal C57BL/6J P3-P7 pups were used for OPC and brain slice preparations, which were combined with natural Treg were isolated exclusively from young (3–4 m) and aged (15–18 m) C57BL6/J male mice for use with neonatal mixed sex pups. For in vivo Treg adoptive transfer studies, natural Treg were isolated from young (3–4 m) and

**Fig. 8 | *Mcam* and *Itga2* contribute to Treg-driven OPC differentiation and are downregulated in aged Treg. A** Representative images of immunostaining showing OPC differentiation in co-culture with young Treg in the presence or absence of neutralising antibodies against candidate cell surface mediators (scale bar = 100 μm). **B** Bar graph showing the quantification of OPC differentiation measured by the fold change in percentage of MBP$^+$ cells ($n = 7$ mice, 2 independent experiments, 2-way ANOVA, Dunnett's multiple comparison tests, $F_{raw\ factor(experiment)}$=2.271, $P_{raw\ (experiment)}$=0.0706, $F_{column\ (treatment)}$=14.26, ***$P_{column\ (treatment)}$<0.0001; **$P_{Control\ vs\ Treg\ isotype}$ = 0.0052, $P_{Treg\ vs\ Treg\ isotype}$ = 0.1164, **$P_{Treg\ isotype\ vs\ Treg\ anti-Mcam}$ = 0.0025, *$P_{Treg\ isotype\ vs\ Treg\ anti-Itga2}$ = 0.0160). **C** Bar graph showing the quantification of OPC differentiation measured by the fold change in percentage MBP$^+$ area per well ($n = 7$, 2-way ANOVA, Dunnett's multiple comparison tests, $F_{raw\ factor(experiment)}$=3.058, *$P_{raw\ (experiment)}$=0.0229, $F_{column\ (treatment)}$=6.108, **$P_{column\ (treatment)}$=0.0016; **$P_{Control\ vs\ Treg\ isotype}$ = 0.0011, $P_{Treg\ vs\ Treg\ isotype}$ = 0.6375, *$P_{Treg\ isotype\ vs\ Treg\ anti-Mcam}$ = 0.0102, *$P_{Treg\ isotype\ vs\ Treg\ anti-Itga2}$ = 0.0128). **D** Representative images of immunostaining showing OPC differentiation in neonatal OPCs treated with recombinant ITGA2 (OLIG2 (cyan), CNPase (grey) and MBP (magenta), scale bar = 50 μm). **E** Bar graph showing quantification of the proportion of MBP-expressing oligodendrocytes in rITGA2-treated neonatal OPCs at 3 days in vitro ($n = 7$ mice, 2 independent experiments, Kruskal Wallis test, Dunn's multiple comparison test, KW statistic=4.199, $P_{VH\ vs\ 0.01μg/mL\ ITGA2}$ = 0.8079, $P_{VH\ vs\ 0.1\ μg/mL\ ITGA2}$ > 0.9999, $P_{VH\ vs\ 1\ μg/mL\ ITGA2}$ > 0.9999). **F** Bar graph showing quantification of MBP$^+$ cell area per well in rITGA2-treated neonatal OPCs at 3 days in vitro $n = 7$ mice, 2 independent experiments, 1-way ANOVA, Dunnett's multiple comparison test, $F = 5.336$, $P_{VH\ vs\ 0.01μg/mL\ ITGA2}$ = 0.9920, $P_{VH\ vs\ 0.1μg/mL\ ITGA2}$ = 0.8916, **$P_{VH\ vs\ 1μg/mL\ ITGA2}$ = 0.0080. **G** Sholl analysis showing the number of intersections as a function of the distance from the cell body to determine changes in the morphology of MBP-expressing oligodendrocytes ($n = 7$ mice, 2 independent experiments, graph shows the average of 3 wells per mice and 6–9 cells per well, 2-way ANOVA, $F_{interaction} = 7.143$, ***$P_{interaction} < 0.0001$; $F_{distance} = 38.42$, ***$P_{distance} < 0.0001$; $F_{treatment} = 8.946$, *$P_{treatment} = 0.0113$; Data shown as individual data points and a non-linear regression fitting curve). Data are represented as mean ± SEM. Source data are provided as a Source Data File.

aged (15 m) C57BL6/J female mice or Foxp3-RFP mice, and were injected into both, female and male recipient Foxp3-DTR mice (2–4 m). To deplete Foxp3$^+$ cells in Foxp3-DTR mice, diphtheria toxin was administered as described below. All animal maintenance and experiments were done in accordance with the UK Home Office regulation (Project Licences 2789 and 2894) and were approved by the Queen's University Belfast's Animal Welfare and Ethical Review Committee.

### Flow cytometric analysis of healthy murine CNS
Male C57BL/6 mice (8-week-old and 75 and 95-week-old) were injected i.v. with biotinylated CD45 antibody (30-F11, eBioscience) 3 min prior to euthanasia via a schedule 1 method. Brain, spinal cord and spleen tissues were dissected out and placed individually into 5 mL cold FACS buffer (2% FCS in PBS). Single cell suspensions were prepared from brain and spinal cord tissues as previously described[57]. Briefly, tissues were chopped into small pieces and digested for 30 min at 37 °C with 0.4 mg/mL collagenase D (Sigma-Aldrich), 300 μg/mL hyaluronidase (Sigma-Aldrich) and 40 μg/mL DNase 1 (Sigma-Aldrich) in IMDM supplemented with 2 mM MgCl2, 2 mM CaCl2, 2 mM l-Glutamine (Sigma-Aldrich), 1 mM Sodium Pyruvate (Gibco) 10 mM HEPES (Gibco), 20% FBS and 1X Gentamycin (Abcam), followed by filtration (through 100 μm mesh) and enrichment for leukocytes by gradient centrifugation (40% Percoll GE healthcare, 600 × g, 15 min). To prepare single cell suspensions from spleen tissues, the spleen was mechanically disrupted between two glass slides, filtered through 100 μm mesh and red blood cells were lysed. Splenocytes were counted using a Countess automated cell counter. Following cell preparation, non-specific binding was blocked using 2.4G2 supernatant. Initial staining included eBioscience Fixable Viability Dye e780 (1:4000, eBioscience) and a fluorescence-tagged streptavidin to identify leukocytes which were from contaminating blood using the injected CD45-biotin (1:1000, eBioscience, Clone 30-F11). Next, a 1 h surface stain was performed including antibodies for CD45 (1:500; eBioscience, Clone 30-F11), CD4 (1:500; eBioscience, Clone GK1.5), CD8 (1:1000 Biolegend, Clone 53-6.7), CD62L (1:1000; eBioscience, Clone Mel-14), CD11b (1:500; eBioscience, Clone M1/70), CD44 (1:500; Biolegend, Clone IM7), CD69 (1:100, eBioscience, Clone H1.2F3), CD25 (1:200; eBioscience, Clone PC61) and CD19 (1:250, eBioscience, Clone ID3) Subsequently, cells were fixed and permeabilised with the eBioscience Foxp3 staining kit (eBioscience) and a 1 h intracellular stain for Foxp3 was added (1:100, eBioscience, Clone FJK-16s; Miltenyi, Clone REA788). For brain and spinal cord, the entire sample was acquired. Flow cytometry data were acquired on a Cytek Aurora and data collection performed using SpectroFlo (Cytek).

### Lysolecithin-induced spinal cord demyelination in vivo
Spinal cord demyelination was induced as described previously[27]. In brief, demyelination was induced in the ventral white matter funiculus of the thoracic spinal cord (between vertebrae T11-12 or T12-13) by the injection of 1.2 μL of 1% (w/v) L-α-Lysophosphatidylcholine (Lysolecithin; Sigma-Aldrich) under general anaesthesia. At 7-, 10- or 14-days post lesion (dpl), mice were terminally anaesthetised with intraperitoneal (i.p.) pentobarbital injections and transcardially perfused with ice-cold phosphate buffered saline (PBS) followed by 4% paraformaldehyde (PFA) (Sigma-Aldrich). Spinal cords were dissected and immersed overnight in 4% PFA at 4 °C. Next, spinal cords were cryoprotected with 30% sucrose (Sigma-Aldrich) in PBS for 72 h and snap-frozen in OCT (Tissue-Tek). Frozen spinal cords were cryosectioned at 12 μm thickness and immunostained as described below.

### EAE induction
Young (3–4 m) and aged (15 m) old C57BL/6J female mice were acquired from Charles River Laboratories. Anesthetised mice were immunised with subcutaneous injections of 150 μg of MOG$_{35-55}$ (GenScript) emulsified in 200 μl of complete Freud's adjuvant (CFA) (Incomplete Freund's Adjuvant containing 2.5 mg/mL Mycobacterium tuberculosis H37Ra, both BD Biosciences). At 0- and 2-days post-immunisation, mice were injected with 200 ng of pertussis toxin intraperitoneally (PHZ1174, Thermofisher). EAE clinical signs and weight were monitored daily from 7 days post-immunisation. Mice were euthanised at 36 days post-immunisation by transcardial perfusion with PBS followed by 4% PFA. Spinal cords were dissected and immersed overnight in 4% PFA at 4 °C. Next, spinal cords were cryoprotected with 30% sucrose (Sigma-Aldrich) in PBS for 72 h and snap-frozen in OCT (Tissue-Tek). Frozen spinal cords were cryosectioned at 12 μm thickness and immunostained as described below.

### In vivo Treg expansion
Young (2–3 m) and aged mice (15 m) were intraperitoneally injected with IL-2/anti-IL-2 antibody complexes to expand endogenous Treg[58]. In brief, murine IL-2 (1 μg; Peprotech) was mixed with anti-mouse IL-2 (5 μg; Bioxcell, Clone JES6-1A12) in PBS at 37 °C for 30 min. Each mouse was injected daily for 3 consecutive days with IL-2 (1 μg) and anti-IL-2 (5 μg) in 200 μl of PBS. Adequate Treg expansion was assessed by flow cytometry as described below. Spinal cord demyelination was induced on the day after the third injection of IL-2/anti-IL-2, by injecting lysolecithin into the ventrolateral white matter as described above. Treg expansion was examined by flow cytometry as described below on day 10dpl in aged mice, while OPC differentiation was examined at 7 dpl in young mice and 10 dpl in aged mice.

### In vivo Treg depletion and adoptive transfer of wild type natural Treg
Young (2–4 m) and aged (19–23 m) male and female Foxp3-DTR mice were injected daily i.p. with diphtheria toxin (DT; 0.04 μg/g of body

weight; Sigma, Cat. No. D0564) for 3 days prior to demyelination. To maintain endogenous Treg depletion during remyelination, DT (0.04 μg/g of body weight) was injected i.p. every fourth day. Control animals received 200 μl of saline i.p. Depletion was confirmed at the endpoint by flow cytometric analysis of endogenous GFP$^+$ natural Treg in blood, spleen, and lymph nodes (see below). To adoptively transfer young (2–4 m) and aged (15 m) natural Treg to Treg-depleted mice, we immunomagnetically isolated natural Treg from wild type female mice or Foxp3-RFP male and female mice using a CD4$^+$ and CD25$^+$ Treg isolation kit (STEMCELL Technologies) as described below. For remyelination experiments, mice were injected i.p. with 10$^6$ wild type natural Treg (which are resistant to DT) in the 24 h prior to lysolecithin-induced demyelination, while in the case of adoptive transfer of RFP$^+$ Treg to evaluate MCAM and ITGA2 expression, 4 × 10$^5$ cells per mouse were injected to non-lesioned mice. Reconstitution was evaluated by flow cytometry in lymph nodes at the endpoint (see below).

## OPC isolation and culture

OPCs were isolated from P3-P7 mice based on A2B5 expression. Briefly, pups were culled by an overdose of pentobarbital (200 mg/mL, 20 μl per mouse) and whole brain was dissected and placed in ice-cold Hibernate-A (ThermoFisher Scientific). Dissected brain was minced using scalpels, transferred to a 15 mL tube, and centrifuged for 1 min at 100 g and 4 °C. Hibernate-A medium was aspirated, and the pellet was resuspended in 5 mL dissociation media containing 165U of Papain (Worthington) and DNAse type I (40 μg/mL; Worthington) in hibernate A for 30 min at 37 °C. Upon digestion papain was washed off with Hanks Buffered Salt Solution (HBSS) (ThermoFisher Scientific) and cells were centrifuged for 5 min at 300 g and 4 °C. The pellet was resuspended in 4 mL trituration buffer (Hibernate-A with 2% B27; ThermoFisher Scientific) and 2 mM sodium pyruvate (ThermoFisher Scientific) and gently triturated ten times using a 5 mL pipette. Tissue was left to settle for 2 min at room temperature (RT), and supernatant was transferred to a clean 50 mL tube through a 70 μm strainer (Corning). Remaining tissue was exposed to another 2 mL of trituration buffer and gently triturated using a glass polished pipette. After trituration, tissue was left to settle for 2 min at RT and supernatant was transferred to the tube through the 70 μm strainer. This step was repeated another time with a glass fire-polished pipette of decreased diameter and then with a 1 mL pipette. To further remove debris from the cell suspension, 11.0 mL of 90% Percoll (GE Healthcare) diluted in 10X PBS (ThermoFisher Scientific) was added and topped up to a final volume of 45 mL with Dulbecco's Modified Eagle's medium (DMEM) (ThermoFisher Scientific). The cell suspension was then centrifuged at 800 g for 20 min at 4 °C. Upon centrifugation, the pellet was washed with HBSS and then resuspended in 10 mL magnetic-activated cell sorting (MACS) buffer (Hibernate-A with 2 mM sodium pyruvate, 2% B27, 0.5% BSA; ThermoFisher Scientific), 2 mM EDTA (Sigma-Aldrich) and 10 μg/mL insulin (Sigma-Aldrich), and incubated for 30 min at 37 °C in a 10 cm petri dish coated with BSC1 Griffonia Simplicifolia Lectin (BSL1, 5 μg/mL)(Vektor Labs) to remove microglia. Then, the supernatant was collected, and cells were counted and centrifuged at 300 g for 5 min at 4 °C. The pellet was resuspended and incubated with 2 μg of A2B5 antibody (Millipore, Clone A2B5-105) in 500 μL of MACS buffer per 10 million cells for 25 min on ice, with gentle resuspensions every 10 min. Cells were washed with HBSS and spun at 300 g for 5 min at 4 °C (Miltenyi Biotec). The pellet was resuspended and incubated with 80 μL of MACS buffer and 20 μL of anti-IgM microbeads (Miltenyi Biotec) for 15 min. Then, cell suspensions were centrifuged at 300 g for 5 min at 4 °C, resuspended in 0.5 ml MACS buffer and placed in a MACS mini column (Miltenyi Biotec) on a MiniMACS Separator (Miltenyi Biotec) and washed with 1.5 mL MACS buffer. Once the liquid passed completely through the column, the column was removed from the stand and placed in a new 15 mL tube, where 1 mL of OPC media was added to the column and plunged through to elute the OPCs. OPC

media contained DMEM, 2% B27, sodium pyruvate (2 mM), insulin (5 μg/mL), Trace Elements B (0.01%; Corning), Forskolin (5 μM; Sigma-Aldrich), Biotin (10 ng/mL; Sigma-Aldrich), Penicillin-streptomycin-glutamine (1%; ThermoFisher Scientific), N-acetyl cysteine (60 μg/mL; Sigma-Aldrich) and 1% SATO stock solution. SATO stock solution contained bovine serum albumin (BSA) fraction V (0.1 mg/mL; ThermoFisher Scientific), sodium selenite (4 μg/mL; Sigma-Aldrich), putrescine (1.61 mg/mL; Sigma-Aldrich), apo-transferrin (0.1 mg/mL; Sigma-Aldrich) and progesterone (4 μg/mL; Sigma-Aldrich). The purified population was counted and diluted in OPC media for plating. OPCs were plated at a density of 3000 cells per well in 96-well plates (Falcon) previously coated with poly-L-lysine (10 μg/mL; Sigma-Aldrich) and laminin (10 μg/mL; Sigma-Aldrich) diluted in DMEM. OPCs were plated with PDGFaa (20 ng/mL; Peprotech) and NT-3 (10 ng/mL; Peprotech). The following day two thirds of the media was removed (100 μl) and 100 μl of fresh OPC media with PDGFaa (20 ng/mL) and NT3 (10 ng/mL) were added per well. Cells were incubated at 37 °C and 5% CO$_2$ for two days. On the third day, media was completely removed and 20,000 Treg in fresh media were added as indicated below.

For transwell assays, OPCs were selected with anti-PDGFRα panning. After microglia depletion in BSL1-coated petri dishes, the supernatant was transferred to PDGFRα panning plates and incubated at RT for 1 h. PDGFRα panning plates were coated with goat-anti-rat IgG (H + L) (7.5 μg/mL; ThermoFisher Scientific) in PBS with 0.2% BSA and rat anti-PDGFRα (1.5 μg/mL in PBS; BD Biosciences) with 0.2% BSA for 3 h. Supernatants were removed, cells were washed 5-8 times with PBS and the attached OPCs were scraped, counted, and replated in 24-well plates (Sarstedt) at 10,000 cells per well. On the third day in vitro 50,000 Treg were added either directly into the well with OPCs or in a transwell (Millipore) above the OPCs.

For recombinant ITGA2 and MCAM treatment, OPC were treated with increasing concentrations of the recombinant proteins in OPC media, or vehicle control, on the day of growth factor removal. Cells were cultured for 3 days in the case of ITGA2 and 6 days in the case of MCAM without any further media changes.

At the endpoint (3 or 6 days after treatment) OPC were fixed for 15 min with 4% PFA and washed twice with PBS prior to immunostaining. Cells were blocked with 5% donkey serum (Sigma-Aldrich) with 0.1% Triton-X-100 (Sigma-Aldrich) in PBS for 1 h at RT. OPCs were then stained overnight at 4 °C with primary antibodies diluted in blocking solution against Olig2 (1:500, Bio-techne), CNPase (1:500, Sigma-Aldrich, clone 11-5B), MBP (1:500, Millipore, clone 12), Ki67 (1:300, Abcam, clone SP6), NG2 (1:200, Millipore), and CD3 (1:500, eBioscience). Cells were then washed three times with PBS and incubated for 1 h at room temperature with secondary antibodies diluted in blocking buffer Alexa fluor (AF) 488 donkey anti-rabbit (1:500; ThermoFisher Scientific), AF568 donkey-anti-rat (1:500; Abcam), AF647 donkey anti-mouse (1:500; Abcam), AF647 donkey anti-rabbit (1:500; ThermoFisher Scientific) and AF755 donkey anti-goat (1:500; ThermoFisher Scientific) and Höechst (1:10,000; Sigma-Aldrich). Cells were then washed with PBS three times and stored in 150 μL PBS/well for cell imaging.

Cell imaging was performed using the CellInsight CX5 high content imaging system (ThermoFisher Scientific) for 96-plates and EVOS for 24-well plates. Twenty-five separate fields of view of each well in 96-well plates and the cell populations of interest were quantified using CellInsight CX5 analysis software and the SpotCounts programme. For 24-well plates we imaged randomly 5 fields of view per well at 20X magnification and images were analysed manually with Fiji.

## Natural treg isolation

Young (2–4 m) and aged (15–18 m) male mice were culled by CO$_2$ overdose. Spleens and lymph nodes were removed and mashed through a 70 μm strainer with a 2 ml syringe plunger. CD4$^+$ T cells were immunomagnetically purified by negative selection according to

manufacturer's instructions (STEMCELL Technologies). Isolated CD4+ T cells were then subjected to a CD25+ cell isolation (STEMCELl Technologies) using releasable magnetic spheres (STEMCELL Technologies) according to the manufacturer's protocol. Purity was confirmed by flow cytometry for CD4, CD25 and Foxp3 (Supplementary Fig. 2A). For i.p. injections in adoptive transfer experiments, cells were resuspended at $10^6$ cells per 200 μL in saline. For OPC-natural Treg co-cultures 20,000 natural Treg were resuspended in 150 μl of OPC Brainphys-based media and added to OPCs in 96-well plates. For 24-well transwell plate experiments 50,000 natural Treg were resuspended in 250 μl OPC-BrainPhys media and added to transwells above cultured OPCs. OPC BrainPhys-based media contains BrainPhys (STEMCELL Technologies), B27 (2%; ThermoFisher Scientific), Glutamax (0.5 mM; ThermoFisher Scientific), N-acetyl-cysteine (60 μg/mL), SATO (1:100, as described above) and insulin (5 μg/mL). In the case of brain slices, 50,000 Treg were diluted in 5 μL of brain slice media and dropped directly onto brain slices.

For natural Treg mechanistic studies, OPCs and Treg were isolated as described above. Prior to OPC-natural Treg co-culture, 120,000 Treg were transferred to a 1.5 mL microcentrifuge tube and centrifuged at 400 g for 5 min at 4 °C. Treg were resuspended in 100 μL of PBS with 2% FCS and a blocking antibody (20 μg/mL) or corresponding isotype were added per tube. Natural Treg were incubated with the blocking antibody for 45 min on ice and then cells were washed with PBS and centrifuged for 10 min at 400 g and 4 °C. Cells were resuspended in 900 μl of OPC BrainPhys-based media (see above) and 20,000 Treg in 150 μL of media were added per well in 96-well plates. As previously, Treg were co-cultured with OPCs for 7 days in vitro (DIV) and then fixed and stained as described above. Blocking antibodies used were anti-NK1.1 PE (eBioscience, clone PK136), anti-CD62L-APC (eBioscience, clone Mel14), anti-Ly6c1-APC (eBioscience, clone RB6-BC5), rat-isotype-APC (eBioscience), anti-Itga2 (Abcam, clone EPR5788), anti-MCAM (Abcam, clone EPR3208) and rabbit-isotype (Vektor labs).

## Organotypic brain slice cultures

Brainstem slices from male and female P3 C57BL/6 J mice were prepared at 300 μm thickness using a McIlwain Tissue Chopper as described previously[27]. Slices were separated and placed on Millicell inserts in individual wells of a 24-well plate (Millipore) with 250 μl brain slice medium containing 46.6% minimum essential medium (ThermoFisher Scientific), 25% Earls balanced salt solution (Sigma-Aldrich), 25% heat-inactivated horse serum (Thermofisher Scientific), 1% penicillin-streptomycin (Thermofisher Scientific), 1% glutamax (Thermofisher Scientific) and 1.4% D-glucose (Sigma). Brain slices were incubated at 37 °C and 5% $CO_2$ overnight and then media was fully replaced. On day 3, media was changed again and 50000 young or aged Treg were added directly on top of each slice in a 5 μl droplet. Brain slices were cultured for a further 7 DIV, with media changes every other day. On day 10 (7 days post-treatment), brain slices were fixed with 4% PFA for 45 min and immunostained. In brief, brain slices were blocked for 2 h in 10% normal donkey serum (Sigma-Aldrich), 1 mM Hepes (Thermofisher Scientific), 1% BSA (ThermoFisher Scientific) and 0.5% Triton X-100 (Sigma-Aldrich) in PBS. Slices were incubated with primary antibodies for rat anti-MBP (1:500; Millipore, clone 12) and chicken anti-NFH (1:500; EncorBiotech, polyclonal) in blocking buffer at 4 °C for two overnights. Brain slices were then washed with PBS-0.01% triton three times for 30 min. Slices were incubated with secondary antibodies AF488 donkey anti-chicken IgY (1:500; Abcam, polyclonal) and AF568 donkey anti-rat IgG (1:500; Abcam, polyclonal) in blocking buffer overnight at 4 °C. Brain slices were washed twice with PBS-0.01% triton and then incubated with Höechst stain (1:20000; Sigma-Aldrich) for 10 min. Slices were washed with PBS and mounted using fluoromount G (Thermofisher Scientific). Imaging was performed using a Leica SP8 confocal microscope, using a 63X oil objective and the Leica Navigator at 0.5 μm intervals over 10 μm. Four fields of view per brain slice were selected for imaging based on NFH and Höechst staining but blinded to MBP. To quantify myelination index, the area stained by NFH, MBP and the colocalising area of NFH+MBP+ were measured per stack in Fiji[59] using a Fiji plug-in developed and kindly provided by the Williams laboratory at the University of Edinburgh[60]. Then, the ratio between total NFH+ and NFH+MBP+ areas (myelination index) was calculated per z-stack and the average of the myelination index per slice was calculated.

## Flow cytometry

Spleens and lymph nodes were mashed through a 70 μm strainer. For splenocytes, cells were exposed to red blood cell lysis buffer (STEMCELL Technologies) for 2 min at room temperature. Both, lymph nodes and splenocytes were then washed with PBS and centrifuged at 300 g for 5 min at 4 °C. Cells were resuspended in 200 μL PBS and stained with a cell viability dye with eFluor 506-UV viability dye (1:2000; ThermoFisher Scientific) and cell surface stained with antibodies for CD4 (1:500; eBioscience, clone RM4.5), CD25 (1:500; eBioscience, clone PC61.5), ITGA2 (1:50; eBioscience, Clone DX5) and MCAM (1:100; Biolegend, Clone ME9F1) for 15 min at RT. Cells were washed with flow cytometry staining buffer (FCSB) (2% FCS in PBS) and centrifuged at 300 g for 5 min at 4 °C. Cells were then fixed with Fix & Perm A (ThermoFisher Scientific) for 10 min at RT. Fixative was washed off with FCSB and centrifuged at 300 g for 5 min at 4 °C. The pellet was resuspended in PBS and data were acquired on a FACSCanto II or on a FACS Symphony in the case of RFP+ adoptive transfer. In the natural Treg depletion experiment, presence of endogenous natural Treg was determined by the expression of GFP. To determine the purity of natural Treg isolations or the extent of natural Treg reconstitution by adoptive transfer, cells were washed and centrifuged for 5 min at 300 g and 4 °C. Cells were then resuspended in 100 μL Fix & Perm B (ThermoFisher Scientific) with an anti-Foxp3 antibody (1:100; eBioscience, clone FJK-16S) overnight at 4 °C. Cells were then washed with FCSB and centrifuged at 300 g and 4 °C for 5 min. Cells were then resuspended, data were acquired on a FACSCanto II and analysed using FlowJo software version 9.0 and 10.0 (BD). To calculate cell numbers, singlets were identified by FSC-H versus FSC-A and viable cells gated for CD3 and CD4, and subsequently CD25 and Foxp3+GFP+ and Foxp3+GFP- cells. To evaluate young and aged Treg numbers in blood, 30 μL were incubated with 100 μL of the viability and cell surface antibody mix for 30 min at room temperature. Cells were then washed with FCSB and centrifuged at 300 g for 5 min at 4 °C. After resuspension, cells were fixed and lysed in 100 μL of OptiLyse B (Beckman Coulter) for 10 min at RT, washed with distilled water and then after 15 min, acquired on a FACSCanto II. After initial data acquisition, cells were centrifuged at 300 g for 5 min at 4 °C and incubated overnight with anti-Foxp3 antibody in Fix & Perm B, as described above. Cells were washed and data were acquired on a FACSCanto II. To evaluate changes in MCAM and ITGA2 expression upon reconstitution, splenocytes were stained as described above but without Fix & Perm A or B, to avoid quenching endogenous RFP and GFP signal and data were acquired from live cells on a BD FACS Symphony. To calculate cell proportions and mean fluorescence intensity, singlets were identified by FSC-H versus FSC-A and viable cells gated for CD4, and subsequently RFP+ GFP- Treg. ITGA2 and MCAM expression based on median fluorescent intensity was then evaluated in RFP+ and ITGA or MCAM+ RFP+ Treg respectively.

## Immunofluorescence staining of CNS tissue

Spinal cord sections were dried for 30 min at RT and washed for 10 min in PBS. For Olig2, Ki67, CC1 and ASPA staining, spinal cord sections underwent antigen retrieval at 85 °C for 10 min with 1X citrate buffer pH 6.0 (Sigma-Aldrich). Tissue sections were washed with PBS and permeabilised with 1% Triton-X-100 in PBS for 30 min at RT. Sections were washed with PBS and incubated with blocking solution (5%

donkey serum; Sigma-Aldrich) diluted in TBS with 0.25% tween (Sigma-Aldrich) for 1 h at RT. If spinal cord sections were incubated with a primary antibody raised in mouse (e.g., anti-CC1) sections underwent an additional 1 h blocking step at RT using Mouse on Mouse (M.O.M) blocking reagent (Vector labs) in 5% donkey serum. Then spinal cord sections were incubated with primary antibodies against Olig2 (1:500; Bio-Techne, polyclonal), anti-APC (1:400, Abcam, clone CC1), anti-ASPA (1:300, Millipore, polyclonal), anti-NFH (1:500; Abcam, polyclonal), anti-MBP (1:500; Millipore, clone 12) or anti-Ki67 (1:300; Abcam, clone SP6) overnight at 4 °C. Sections were washed with Tris-Buffered-Saline (TBS) (ThermoFisher Scientific) with 0.25% Tween (Sigma-Aldrich). Secondary antibodies including donkey anti-goat AF488, donkey anti-mouse AF568, donkey anti-rabbit AF647 (all 1:500; ThermoFisher Scientific), donkey anti-rat AF568 (1:500; Abcam), donkey anti-rabbit AF647 (1:500, Abcam) and Höechst (1:10000; Sigma-Aldrich) were added for 1 h at room temperature. Spinal cord sections were then washed with TBS-0.25% Tween twice and mounted with fluoromount G.

For detection of CD4 and Foxp3, spinal cord sections underwent antigen retrieval at 80 °C for 5 min. After blocking with 10% NDS, sections were incubated with rat-anti-mouse CD4 (1:500; Biolegend, Clone RM4.5) and rabbit-anti-mouse Foxp3 (1:500; Biotechne, polyclonal) for 1 h at room temperature, washed and incubated with secondary antibodies as described above.

Image acquisition was performed using the Leica TIRF and Leica DM5500 widefield fluorescent microscopes. Further image processing was performed in Fiji[59] software and analysis was undertaken by blinded manual counting. For NFH and MBP wrapping analysis, images were quantified using Cell Profiler and Cell Profiler Analyst softwares[61,62].

### RNA sequencing and analysis

Young (2–4 m) and aged (16 m–20 m) natural Treg were isolated from spleen and lymph nodes by magnetic-activated cell sorting and purity was checked by flow cytometry as described above (Supplementary Fig. 3A). Natural Treg were lysed in 0.5 mL of Trizol (Sigma-Aldrich) by vortexing and pipetting and frozen at −80 °C. RNA was extracted combining Trizol and Chloform isolations with RNAeasy micro kit columns (Qiagen). RNA concentration was determined using a Nanodrop, and RNA quality was assessed by Qubit measurement and in an RNA nanochip Bioanalyzer. Sequencing libraries were prepared using the Kapa Hyper Preparation kit with riboerase (Roche) following manufacturer's instructions. Sequencing was performed on the NovaSeq 6000 in a pair-end 75 base pair format and 80 million reads.

The raw sequencing data was checked for lower quality bases and adaptor sequences with FastQC and quality trimming was performed using CutAdapt[63]. The high-quality trimmed data were mapped on 10 mm mouse reference genome using STAR aligner (v 2.7.0a)[64]. The quantification of each gene transcripts was performed with ensemble transcript annotation gtf file (GRCm38) using featureCounts tool (v 2.0.0)[65]. The resultant read counts matrix was analyse using DESeq2 (v1.38.1)[66] to identify differentially expressed genes between the young and aged natural Treg samples using Wald tests with the Bonferroni correction for multiple testing. The differentially expressed genes (DEG) with a multiple testing adjusted $p < 0.05$ and fold change $>2$ were considered for further downstream analysis. These genes were subjected to gene ontology enrichment analysis using enrichGO function in clusterProfiler (v4.2) R package and DAVID (https://david.ncifcrf.gov). Over and underrepresented ontology terms were identified ($p <= 0.05$ and Bonferroni correction p adjustment method). Next, enriched activated and suppressed functional pathways were identified using gseKEGG function in clusterProfiler with $p < 0.05$.

Raw data files from RNAseq are available at the GEO database (accession number GSE218804).

### Bioinformatic analysis to identify putative Treg mechanisms

To determine the molecular mechanisms associated with Treg-driven OPC differentiation that are impaired with ageing, we combined our young and aged natural Treg RNA sequencing with data obtained from Zhang et al. 2014 (GSE: 52564)[45]. We first extracted genes that were downregulated in aged natural Treg and associated with the GO cellular component term "plasma membrane". We then extracted the FPKM matrix and selected genes that were upregulated in OPC when compared with newly myelinating oligodendrocytes and oligodendrocytes in the Zhang et al. database. We performed GO cellular component analysis on those OPC-enriched genes and selected genes associated with the GO term Apical Plasma Membrane. Then, both gene lists were subjected to a protein-protein interaction analysis using the String open-source tool (www.string-db.org)[46]. Protein–protein interactions within genes that were associated only with the natural Treg or only the OPC databases were eliminated, and the remaining downregulated natural Treg membrane proteins were prioritised considering their significance, fold change and the number of potential interactors enriched in OPCs. Candidate genes were then selected based on their p-adjusted value, fold enrichment and the number of interactions with OPC receptors.

### Statistical analysis

All statistical analyses were performed in GraphPad Prism (GraphPad Software, Inc. version 9) or R. First normality of datasets was assessed using Shapiro-Wilk and Kolmogorov-Smirnov tests. For comparisons between 2 groups, such as young and aged Treg and OPC differentiation (Supplementary Fig. 1), unpaired two-tailed Student's t tests were used if data were normally distributed, and Mann-Whitney U tests for non-parametric datasets. When more than two groups were analysed and datasets were normally distributed, a one-way analysis of variance (ANOVA) was performed assuming equal variances, followed by Sidak's multiple comparison test. When two factors were controlling the outcome, such as replicates and treatments, 2-way ANOVA followed by Dunnet's comparison was used. When datasets were was not normally distributed, a Kruskal-Wallis followed by Dunn's multiple comparison test was used. For percentage data, *arcsin* conversion was performed to analyse the data using parametric tests. For all statistical tests, differences were considered significant at $p < 0.05$. Outlier test was run on Prism in every file and any value identified as a statistical outlier removed from the graph. Additionally, in the in vivo experiment animals that were either not depleted (>3% GFP+ cells either in blood, spleen or lymph nodes) or not successfully reconstituted (>3% Foxp3+ CD4+ cells) upon young and aged adoptive transfer were removed from subsequent analysis except for the aged Foxp3-DTR groups in which we show no reconstitution at all neither in spleen nor lymph nodes as indicated in results. Animals which lesions were below 0.045 μm were also removed from the analysis as small lesions have a higher regenerative potential and that may induce biased in the final outcome.

### Reporting summary

Further information on research design is available in the Nature Portfolio Reporting Summary linked to this article.

## Data availability

Young and aged natural Treg RNA sequencing data generated in this study are available at GEO (accession number: GSE218804). Source data are provided with this paper.

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

## Acknowledgements

We acknowledge extensive technical support from Carmel McVicar, Maria P Athanasios, Michelle Naughton and the staff of the animal facility. We thank A. Rudensky (Memorial Sloan Kettering Cancer Centre) for providing Foxp3-DTR mice. We thank the laboratory of Anna Williams (University of Edinburgh) for the Fiji plug-in to analyse myelination index in brain slice cultures.

This work was supported by the Wellcome Trust (110138/Z/15/Z to D.C.F.), Biotechnology and Biological Sciences Research Council (BB/J01026X/1 and BB/N003721/1, to D.C.F.), ECTRIMS postdoctoral fellowship (to A.G.F.), Wellcome ISSF fellowship (to A.G.F.), Miguel Servet Fellowship from the Spanish Institute of Health Carlos III (CP21/00032 to A.G.F.), Spanish State Research Agency Plan Generación de Conocimiento 2021 grant (PID2021-124465OA-I00 to A.G.F.), The Leverhulme Trust (ECF-2014-390, to Y.D.), a postdoctoral fellowship contract from the Generalitat Valenciana APOSTD21 (to F.J.R) and postgraduate studentship support from the Dept. for the Economy (Northern Ireland).

## Author contributions

Experiments were designed by A.G.F., D.C.F., A.L., R.J.M, J.D., Y.D. and V.T. Experiments were performed and/or analysed by A.G.F., M.D., E.H., N.V.G., J.W., A.Y., K.M., J.F., S.C.F., T.M., A.D., F.J.R.B., K.S.R., C.E.M., M.I., J.D., R.P., Y.D. and D.C.F. A.L, R.J.I, R.J.M and C.E.M. provided advice on experimental design and interpretation. Manuscript was written by A.G.F. and D.C.F. with contributions from all authors. D.C.F and A.G.F. oversaw the study.

## Competing interests

The authors declare no competing interests.

## Additional information

[1]Wellcome-Wolfson Institute for Experimental Medicine, Queen's University Belfast, Belfast BT9 7BL Northern Ireland, UK. [2]Institute for Health and Biomedical Sciences of Alicante (ISABIAL), Alicante 03010, Spain. [3]Instituto de Neurociencias CSIC-UMH, San Juan de Alicante, Alicante 03550, Spain. [4]Department of Pathology, University of Cambridge, CB2 1QP Cambridge, UK. [5]Babraham Institute, CB22 3AT Cambridge, UK. [6]Department of Clinical Neurosciences,

University of Cambridge, Cambridge, UK. [7]Department of Medical Epidemiology and Biostatistics, Karolinska Institute, Stockholm, Sweden. [8]Institute of Molecular Medicine, University of Southern Denmark, 5000 Odense, Denmark. [9]Danish Institute for Advanced Study (DIAS), 5230 Odense, Denmark. [10]Altos Labs - Cambridge Institute of Science, Granta Park, Cambridge CB21 6GP, UK. [11]Department of Clinical Genetics, Odense University Hospital, 5000 Odense, Denmark. [12]Present address: Division of Pharmacology, Institute for Pharmaceutical Sciences, Utrecht University, Utrecht, The Netherlands. [13]Present address: Patrick G Johnston Centre for Cancer Research, Queen's University Belfast, Belfast BT9 7BL Northern Ireland, UK. [14]Present address: CRUK Beatson Institute, G61 1BD Glasgow, UK. ✉e-mail: guzman_ale@isabial.es; d.fitzgerald@qub.ac.uk

