## [Peer Review File · Nature Communications]

Ageing impairs the regenerative capacity of regulatory T cells in central nervous system remyelinationREVIEWER COMMENTS

Reviewer #1 (Remarks to the Author):

In their manuscript Guzman de la Fuente compare the effects of Tregs from young and old mice on mouse oligodendrocytes. Expansion of Treg in aged mice was not associated with improved or impaired remyelination capacity in old mice. However, they observed a lower capacity of young compared to old Treg to promote differentiation of primary neonatal oligodendrocytes and myelination in cerebellar slice cultures *in vitro*. They found no differences in the effects of intraperitoneally injected Treg from young or old mice on remyelination in lysolecithin induced demyelination (after depletion of endogenous Treg). Aging of Treg was associated with a significantly changed transcriptome. To determine whether contact mediated mechanisms or soluble factors contribute to the pro-regenerative effect of young Treg, the authors added Tregs either directly or in a transwell approach to neonatal oligodendrocytes. Only direct interaction of Treg and OPC was associated with an increased oligodendroglial differentiation. Using bioinformatics methods the authors identified several potential receptor-ligand interactions between OPC and Treg. Anti-ITAG2 and Anti-MCAM were able to reduce the beneficial effect of Treg on oligodendroglial differentiation.

The authors conclude "our findings demonstrate that ageing limits the neuroregenerative capacity of Treg, likely limiting their remyelinating therapeutic potential in aged patients and describe two novel mechanisms implicated in Treg-driven remyelination that may be targetable to overcome this limitation".

This could be in principle an interesting story, however there are several issues that limit my enthusiasm for this manuscript.

1. I do not agree with the conclusion of the authors. The *in vivo* data indicate that aging of Treg has neither a convincingly beneficial nor detrimental effect on oligodendroglial differentiation and/or remyelination *in vivo*. Expansion of Tregs *in vivo* had no significant effect on oligodendroglial differentiation compared to control (why did the authors did not analyze the extent of remyelination in these mice?). However, this does not have to be the consequence of a reduced regenerative potential of Treg, but can be easily explained by an CNS environment which is less responsive to pro-remyelinating cues. Interestingly, there were no significant differences between the number of mature oligodendrocytes or the remyelination index after injection of either young or old Tregs in the lysolecithin model in young mice demonstrating that old Tregs are not less efficient in the promotion of remyelination. If the authors want to convince the reader that aged Treg are less neuroregenerative than young Treg, they should repeat this experiment in aged mice. If young, but not old Treg can rescue the impaired remyelination in old mice, the statement of the authors would become much more convincing.

2. The authors try very hard to stress the difference regarding the effects of young and old Treg on oligodendrocytes *in vitro*. However, in most experiments there are no significant differences between young and old Treg, when you compare these two groups directly. In Fig 2E the apparent (but statistically not significant) difference is due to three outliers in the young Treg group and in Fig 2G due to one outlier in the old Treg group. Again, it would be very helpful to see, whether either young or old Treg are able to overcome the impaired differentiation of old primary oligodendrocytes to support the statement of the authors that aging limits the neuroregenerative effect of Treg.

3. Based on the results of the experiments so far, I do not see the relevance of the findings for a therapeutic approach. First of all, the observed effects of the blocking antibodies are not significant compared to the relevant control (namely the isotype control). Furthermore, the authors do not demonstrate a beneficial effect of either recombinant ITAG2 or NCAM or overexpression of these two factors on oligodendroglial differentiation, which in my opinion should be the first step to determine whether these two factors could serve as a therapeutic target.

Minor issue:

In the introduction and discussion the author state that remyelination is neuroprotective and declines with age. This has been shown in mice, however data supporting the relevance of remyelination for clinical recovery and neuroprotection as well as the decline of remyelination with aging is scarce in MS patients. Therefore, the authors should state more clearly whether they base their statements on findings from animal or human studies.

Reviewer #2 (Remarks to the Author):

Review of de la Fuente et al:

In this manuscript, the authors report that regulatory T cells (Tregs) from aged mice are impaired in their ability to promote remyelination in the context of a multiple sclerosis mouse model. Moreover, using RNA-Seq, they identify Melanoma Cell Adhesion Molecule 1 (MCAM1) and Integrin alpha 2 (ITGA2) as factors expressed in Tregs derived from young mice that they believe are involved in this age-related failure in the ability of aged Tregs to promote remyelination. Interestingly, they find that transfer of aged Tregs into young mice restores the ability of these Tregs to promote remyelination. More specifically, the authors find that, despite an increased fraction of splenic Tregs, old mice had a reduced fraction of mature (Olig2+CC1+) oligodendrocytes (oligos). This was true even when mice were administered IL-2/anti-IL2 complexes to increase the numbers of Tregs. Additionally, aged Tregs were impaired in their ability to convert OPCs into functional/myelinating oligos in vitro and in organotypic brain-slice cultures. Adoptive transfer of young Tregs into lysolecithin-administered Foxp3-DTR mice treated with DT led to a significant increase in the fraction of Olig2+CC1+ and Olig2+ASPA+ oligos, suggesting that young Tregs were capable of promoting oligo maturation and remyelination. Surprisingly, adoptive transfer of Tregs from old mice were also capable of promoting oligo maturation and repair, leading to the hypothesis that Treg regenerative capacity with respect to aging is influenced by the local environment. To address the age-dependent mechanism involved in Treg-mediated oligo maturation, the authors conducted RNA-Seq between young and aged Tregs and identified a number of transcripts that were differentially expressed, with an unexpected increase in 31 Treg tissue repair hallmark genes. Using Treg/Oligo transwell assays, the authors determined that Treg-mediated oligo maturation was cell-cell contact dependent and used this information to search for ligand/receptor pairs that had decreased expression in aged Tregs. This yielded several hits, including Mcam and Itga2 and blocking Abs against these led to a reduction of MBP+ cells in their in vitro OPC maturation assays.

Taken together, these findings provide important new insight regarding the regenerative capacity of Tregs within the CNS and may lead to novel therapeutics for treatment of diseases such as MS. Moreover, these findings are likely to prompt additional discovery regarding the influence of aging on Treg function, particularly within the CNS, and this could be relevant to other CNS diseases, including Alzheimer's Disease. For this reason, I believe that the manuscript is of high quality and likely to be of broad interest to the readers of Nature Communications. Nevertheless, there are some concerns that should be addressed prior to publication.

Concerns:

1. While it is interesting to know that the authors observed an increase in the proportion of Tregs in the spleens of older mice compared to young mice, it would seem important to know if this difference extends to CNS-resident T cell populations, particularly in the context of EAE and/or lysolecithin-treated demyelinating mouse models.
2. As described above, the authors found that adoptive transfer of aged Tregs into young DT-treated Foxp3-DTR mice led to comparable OPC maturation following lysolecithin lesioning as observed with transfer of young Tregs. They also found that Tregs required cell/cell contact to promote OPC maturation and identified several receptor/ligand genes that had muted expression in aged Tregs and found that Ab blockade of two (MCAM and ITGA2) inhibited this activity. It would seem important to evaluate the expression of these two (and other) identified receptor/ligand genes in young vs. old Treg ablated lysolecithin-treated mice to determine if transfer of aged Tregs into a young mouse restores expression of these factors.
3. Does in vivo administration of blocking Abs to MCAM and/or ITGA2 prevent OPC maturation and

remyelination following lysolecithin lesioning? What is the impact on demyelination in MOG35-55 EAE?

4. Little information is provided regarding the mechanism through which aged Tregs have a reinvigorated ability to promote OPC maturation and remyelination following transfer into lysolecithin lesioned young mice. For example, does transfer of young Tregs into aged Treg-depleted and lesioned mice result in a loss of OPC maturation, etc? Does this reinvigoration mechanism require cell-cell contact? It seems this could be fairly easily worked out using their cerebellar slice co-cultures.

Reviewer #3 (Remarks to the Author):

This manuscript is about the role of T-reg in CNS myelin repair. It has been shown previously that T-reg from young animals can promote remyelination. Here, the team is studying the impact of aging on this specific T-reg function. The topic is timely and exciting; however, the team needs to address issues to strengthen the results.

The main concern is related to the location of the T-reg, considering that contact is required to promote differentiation in vitro (Figure 5). The team must show that they are in the CNS in close proximity to OPCs to promote remyelination

Figure 1: the team shows that cells expansion in old mice does not promote myelin repair using the lysolecithin model. The team should show baseline count for oligodendrocyte (panel E and I) as age probably impact these numbers.

The authors should present the data showing that expansion of T-reg in young animals promote remyelination to allow the readers to understand the data better; this piece is critical since authors shows in figure 3 that transfer of old T-reg in a young mice promote myelin repair. Further the authors show expansion of T-reg in the periphery but what about the CNS (meninges and parenchyma).

Figure 2: Line 84 and the paragraph need to be edited, authors only look at MBP expression in vitro and myelin development in slices – not myelin regeneration!

Figure 3 to 5: Does the transcriptome of young and old T-reg injected in T-reg depleted young animals equalize. Showing no difference in expression of MCAM and ITGA2 by qpcr and or flow cytometry would prove that the in vitro mechanism is possible in vivo

While the authors indicate the number of mice used for each panel (n), they do not include the number of independent replicate (N). Each key result should have an N of 2.

REVIEWER COMMENTS

Reviewer #1 (Remarks to the Author):

In their manuscript Guzman de la Fuente compare the effects of Tregs from young and old mice on mouse oligodendrocytes. Expansion of Treg in aged mice was not associated with improved or impaired remyelination capacity in old mice. However, they observed a lower capacity of young compared to old Treg to promote differentiation of primary neonatal oligodendrocytes and myelination in cerebellar slice cultures *in vitro*. They found no differences in the effects of intraperitoneally injected Treg from young or old mice on remyelination in lysolecithin induced demyelination (after depletion of endogenous Treg). Aging of Treg was associated with a significantly changed transcriptome. To determine whether contact mediated mechanisms or soluble factors contribute to the pro-regenerative effect of young Treg, the authors added Tregs either directly or in an transwell approach to neonatal oligodendrocytes. Only direct interaction of Treg and OPC was associated with an increased oligodendroglial differentiation. Using bioinformatics methods the authors identified several potential receptor-ligand interactions between OPC and Treg. Anti-ITAG2 and Anti-MCAM were able to reduce the beneficial effect of Treg on oligodendroglial differentiation. The authors conclude “our findings demonstrate that ageing limits the neuroregenerative capacity of Treg, likely limiting their remyelinating therapeutic potential in aged patients and describe two novel mechanisms implicated in Treg-driven remyelination that may be targetable to overcome this limitation”.

This could be in principle an interesting story, however there are several issues that limit my enthusiasm for this manuscript.

We thank the reviewer for the precise summary and appreciation of the relevance of our work. We have addressed his/her comments point-by-point below.

1. I do not agree with the conclusion of the authors. The *in vivo* data indicate that aging of Treg has neither a convincingly beneficial nor detrimental effect on oligodendroglial differentiation and/or remyelination *in vivo*. Expansion of Tregs *in vivo* had no significant effect on oligodendroglial differentiation compared to control (why did the authors did not analyze the extent of remyelination in these mice?). However, this does not have to be the consequence of a reduced regenerative potential of Treg, but can be easily explained by an CNS environment which is less responsive to pro-remyelinating cues. Interestingly, there were no significant differences between the number of mature oligodendrocytes or the remyelination index after injection of either young or old Tregs in the lysolecithin model in young mice demonstrating that old Tregs are not less efficient in the promotion of remyelination. If the authors want to convince the reader that aged Treg are less neuroregenerative than young Treg, they should repeat this experiment in aged mice. If young, but not old Treg can rescue the impaired remyelination in old mice, the statement of the authors would become much more convincing.

We thank the reviewer for this comment. Based on our results, we believe, that since *in vitro*, aged Treg are not capable of driving OPC differentiation (**Fig. 3C-D**), the failure of *in vivo* Treg expansion to boost regeneration in the aged CNS is a combination of impaired intrinsic Treg regenerative capacity, known CNS-intrinsic limitations in regenerative capacity (e.g. OPC responsiveness to CNS-derived pro-differentiation cues, impaired microglia and macrophage-mediated myelin debris clearance) and as suggested by the reviewer a refractory aged CNS environment that does not respond to Treg-derived cues. We have now included this in our discussion.

As suggested by the reviewer, we have assessed remyelination in the aged CNS with and without peripheral Treg expansion by staining for NFH and MBP colocalization as a surrogate indicator of myelin wrapping around axons. In alignment with the lack of effect of Treg expansion on OPC differentiation, neither is remyelination improved by Treg expansion in this aged model (new data in **Fig. 2 J-L**).

We also performed the suggested *in vivo* adoptive transfer experiment of Treg-depleted aged Foxp3DTR mice receiving young or aged Treg (**Fig. 5**). Surprisingly, in aged mice, OPC survival or migration is also impaired by the absence of Treg as indicated by a decreased density of Olig2⁺ cells (**Fig. 5 A-D**), and this is not rescued by young or aged Treg. However, the decreased density of Olig2⁺CC1⁺ oligodendrocytes in the absence of Treg is partially rescued by transfer of young Treg into these aged animals (no significant difference compared to non-depleted controls) but less so by aged Treg (significantly lower than non-depleted controls). It is important to note that there was no statistically significant difference between groups receiving young and aged Treg (**Fig. 5 B, D**). However, we are cautious interpreting these results in general as we could not find the adoptively transferred Treg in the classic lymphoid organs (spleen or lymph nodes) (**Sup. Fig. 4E, F**), suggesting that the adoptively transferred cells have a different migration pattern with age, and thus we cannot be sure that young and aged cells have the same reconstitution capacity or pattern. We have highlighted these caveats in our results and discussion and stated, that the outcome is likely a consequence of combined impaired Treg regenerative capacity with age and impaired responsiveness of the aged CNS environment.

2. The authors try very hard to stress the difference regarding the effects of young and old Treg on oligodendrocytes *in vitro*. However, in most experiments there are no significant differences between young and old Treg, when you compare these two groups directly. In Fig 2E the apparent (but statistically not significant) difference is due to three outliers in the young Treg group and in Fig 2G due to one outlier in the old Treg group. Again, it would be very helpful to see, whether either young or old Treg are able to overcome the impaired differentiation of old primary oligodendrocytes to support the statement of the authors that aging limits the neuroregenerative effect of Treg.

We appreciate the reviewer's comment in regarding our *in vitro* experiments as we may not have explained our results clearly. In our previous version, our statistical comparisons were performed only against the control group as the questions we sought to address were 'do young Treg alter OL differentiation?' and 'do aged Treg alter OL differentiation?'. As we did not compare changes between young and aged Treg *in vitro*, there were no statistical indications for this comparison. We have now performed this comparison in the datasets from neonatal OPC-Treg co-cultures (**Fig. 3E**) and observed a statistically significant difference in the capacity of young and aged Treg to induce OPC differentiation (based on MBP expression, **Fig 3E**). We also observe a lack of difference between young and aged Treg conditions based on CNPase expression (**Fig 3D**) which suggests that the impairment is most apparent in the later stages of oligodendrocyte differentiation. We have added this to the results and discussion of the revised manuscript.

Following the reviewer's suggestion regarding the potential effect of three outliers in our brain slice experiments, we performed statistical analysis to detect potential outliers and detected only one, the animal that had the lowest myelination index in control which was excluded from the analysis. However, the other data pointed out by the reviewer are not outliers; the spread of the data simply reflects the intrinsic variability of this model. Due to this inherent variability, to increase statistical power, we increased the n number to 9 (donor pups with datapoints averaged from 4 fields of view from 1 brain slice per pup). (**Fig. 3G**). As per our previous data analysis, young Treg induced a statistically significant increase in the myelination index compared to controls, but aged Treg did not. However, in contrast to findings in our OPC culture model, we did not observe a significant statistical

difference between the raw myelination index of young and aged Treg (**Fig. 3G**). Thus, on balance and considering both, the *in vitro* and *ex vivo* data, we interpret that aged Treg do not significantly increase OPC differentiation and are not able to accelerate myelination in brain slices, suggesting that aged Treg are less potent driving OPC differentiation in two models that are receptive to pro-differentiation signals. We have added these results in the revised manuscript.

While we agree that the reviewer's question of whether aged OPCs may be unresponsive to cues from Treg, we have not followed this up for two reasons. First, it is well established that aged OPC do not respond to most pro-differentiation queues (9cRA, T3, Miconazole etc.) as shown by Neuman *et al.*, Cell Stem Cell, 2019. Second, the question our study sought to address was whether ageing impaired the regenerative capacity of Treg and thus, diverging to the effects of ageing on OPC would dilute the focus of the manuscript. Our approach of using neonatal OPCs provides an experimental model that is highly responsive to differentiation cues and is the most direct way to address whether aged Treg have decreased regenerative capacity. That said, we agree with the reviewer that an impairment of regenerative Treg mechanisms would be additive to aged-related OPC differentiation impairment in limiting regeneration in age overall, and we have highlighted this in discussion.

3. Based on the results of the experiments so far, I do not see the relevance of the findings for a therapeutic approach. First of all, the observed effects of the blocking antibodies are not significant compared to the relevant control (namely the isotype control). Furthermore, the authors do not demonstrate a beneficial effect of either recombinant ITGA2 or NCAM or overexpression of these two factors on oligodendroglial differentiation, which in my opinion should be the first step to determine whether these two factors could serve as a therapeutic target.

As indicated above, in the initial manuscript we had exclusively performed post-hoc statistical analysis comparing the data with Treg, but not with the isotype control, which is why no statistical indication was present. We apologise for not explaining this clearly, which may have led to misinterpretations of the results. We have now included this comparison in our post-hoc analysis and demonstrated a significant difference between the isotype and anti-ITGA2 and anti-MCAM, suggesting that indeed the blocking of these interactions significantly impairs pro-regenerative capacity of Treg (**Fig. 8 B, C**).

As suggested by the reviewer, we have now also investigated whether ITGA2 or MCAM directly promote OPC differentiation *in vitro*. We treated neonatal OPCs with recombinant ITGA2 or MCAM and analysed MBP expression (**Fig. 8D-K**). ITGA2 enhanced OPC differentiation into oligodendrocytes, supporting the idea that blocking ITGA2 in young Treg, and reduction of ITGA2 expression by aged Treg, abrogates their pro-differentiation capacity. However, recombinant MCAM did not enhance OPC differentiation, suggesting that Treg-derived MCAM alone is not directly promoting OPC differentiation. It may be that MCAM expression on Treg is necessary for OPC-Treg interaction and/or to trigger the expression of pro-regenerative factors yet to be identified (speculatively this could include, for example, ITGA2 or CCN3 (Dombrowski *et al.*, 2017)) by Treg to drive OPC differentiation. We have included these results in the text and modified our statement regarding the relevance for therapeutic approaches of these findings.

Minor issue:

In the introduction and discussion the author state that remyelination is neuroprotective and declines with age. This has been shown in mice, however data supporting the relevance of remyelination for clinical recovery and neuroprotection as well as the decline of remyelination with aging is scarce in

MS patients. Therefore, the authors should state more clearly whether they base their statements on findings from animal or human studies.

We agree with the reviewer that the neuroprotective role of remyelination has been more widely described in rodents than in humans. However, now there is also evidence in human post-mortem studies and clinical trials of: a) remyelination (Patrikios et al., *Brain*, 2006, Brown et al., *Neuroimage*, 2013; Green et al. *Lance*, 2017; Schwartzbach et al., *J Neurol*, 2017), b) the neuroprotective role of remyelination (Ricigliano et al., *Eur J Neurol*, 2022) and c) the fact that remyelination declines with age (McMurrin et al., *Ann Clin Transl Neurol*, 2022). We have included the corresponding references in the introduction and commented in the discussion.

Reviewer #2 (Remarks to the Author):

Review of de la Fuente et al:

In this manuscript, the authors report that regulatory T cells (Tregs) from aged mice are impaired in their ability to promote remyelination in the context of a multiple sclerosis mouse model. Moreover, using RNA-Seq, they identify Melanoma Cell Adhesion Molecule 1 (MCAM1) and Integrin alpha 2 (ITGA2) as factors expressed in Tregs derived from young mice that they believe are involved in this age-related failure in the ability of aged Tregs to promote remyelination. Interestingly, they find that transfer of aged Tregs into young mice restores the ability of these Tregs to promote remyelination. More specifically, the authors find that, despite an increased fraction of splenic Tregs, old mice had a reduced fraction of mature (Olig2+CC1+) oligodendrocytes (oligos). This was true even when mice were administered IL-2/anti-IL2 complexes to increase the numbers of Tregs. Additionally, aged Tregs were impaired in their ability to convert OPCs into functional/myelinating oligos in vitro and in organotypic brain-slice cultures. Adoptive transfer of young Tregs into lysolecithin-administered Foxp3-DTR mice treated with DT led to a significant increase in the fraction of Olig2+CC1+ and Olig2+ASPA+ oligos, suggesting that young Tregs were capable of promoting oligo maturation and remyelination. Surprisingly, adoptive transfer of Tregs from old mice were also capable of promoting oligo maturation and repair, leading to the hypothesis that Treg regenerative capacity with respect to aging is influenced by the local environment. To address the age-dependent mechanism involved in Treg-mediated oligo maturation, the authors conducted RNA-Seq between young and aged Tregs and identified a number of transcripts that were differentially expressed, with an unexpected increase in 31 Treg tissue repair hallmark genes. Using Treg/Oligo transwell assays, the authors determined that Treg-mediated oligo maturation was cell-cell contact dependent and used this information to search for ligand/receptor pairs that had decreased expression in aged Tregs. This yielded several hits, including Mcam and Itga2 and blocking Abs against these led to a reduction of MBP+ cells in their in vitro OPC maturation assays. Taken together, these findings provide important new insight regarding the regenerative capacity of Tregs within the CNS and may lead to novel therapeutics for treatment of diseases such as MS. Moreover, these findings are likely to prompt additional discovery regarding the influence of aging on Treg function, particularly within the CNS, and this could be relevant to other CNS diseases, including Alzheimer's Disease. For this reason, I believe that the manuscript is of high quality and likely to be of broad interest to the readers of Nature Communications. Nevertheless, there are some concerns that should be addressed prior to publication.

We appreciate the reviewer's positive evaluation of the manuscript and its relevance for the broad audience of Nature Communications.

Concerns:

1. While it is interesting to know that the authors observed an increase in the proportion of Tregs in the spleens of older mice compared to young mice, it would seem important to know if this difference extends to CNS-resident T cell populations, particularly in the context of EAE and/or lysolecithin-treated demyelinating mouse models.

To address this, we investigated whether changes in the CNS-resident T cell populations in healthy brain and spinal cord are present with age. As shown by flow cytometry, there is an increase in the proportion of CD4⁺ cells, CD8⁺ cells, Treg and also tissue resident Treg (CD69⁺) in aged murine CNS tissue with significant differences observed more prominently in spinal cord than brain (**Fig 1F-K, Sup. Fig. 1 A-H**). We also investigated whether ageing affected Treg infiltration in two different models of CNS demyelination (lysolecithin-induced demyelination and EAE) and did not observe statistically significant differences between young and aged CNS at the time points examined (**Fig. 1M, N**). We have included these data in **Fig 1 and Sup. Fig. 1**.

2. As described above, the authors found that adoptive transfer of aged Tregs into young DT-treated Foxp3-DTR mice led to comparable OPC maturation following lysolecithin lesioning as observed with transfer of young Tregs. They also found that Tregs required cell/cell contact to promote OPC maturation and identified several receptor/ligand genes that had muted expression in aged Tregs and found that Ab blockade of two (MCAM and ITGA2) inhibited this activity. It would seem important to evaluate the expression of these two (and other) identified receptor/ligand genes in young vs. old Treg ablated lysolecithin-treated mice to determine if transfer of aged Tregs into a young mouse restores expression of these factors.

As suggested by the reviewer, we have performed flow cytometric analysis of the expression of ITGA2 and MCAM in young and aged RFP⁺ control Treg and young and aged RFP⁺ Treg injected in young Treg depleted Foxp3DTR animals (without lysolecithin lesion). When analysing the Median Fluorescent Intensity (MFI) of RFP⁺ITGA2⁺ or RFP⁺MCAM⁺ CD4 T cells, we observed that indeed aged RFP⁺ cells show a decreased MFI compared to young controls and this was restored upon adoptive transfer into young Treg-depleted mice, although the rescue was more pronounced for ITGA2. Hence, both ITGA2 and MCAM expression appears to be plastic in the aged Treg, being enhanced after adoptive transfer, supporting the idea of a restoration of function. However, due to experimental limitations (e.g. problems with faulty isolation kits) and the limited availability of aged-transgenic Foxp3-RFP mice, we have a very low number in this experiment (n=2-3) which is the reason for the lack of statistical analysis on this result and why it has been included as supplementary data (**Sup. Fig. 6**).

3. Does *in vivo* administration of blocking Abs to MCAM and/or ITGA2 prevent OPC maturation and remyelination following lysolecithin lesioning? What is the impact on demyelination in MOG35-55 EAE?

To address a comment of the previous reviewer, we compared the pro-differentiation capacity of recombinant ITGA2 and MCAM in OPCs *in vitro*. While ITGA2 drove OPC differentiation, this was not observed with MCAM, suggesting that MCAM is involved in OPC-Treg interaction (that likely supports OPC differentiation), but that it does not directly induce OPC differentiation (**Fig. 8**). We have not performed the indicated experiment in EAE because EAE is not ideal for remyelination studies, in part due to significant axonal loss. EAE is an ideal model for investigating the pathogenesis of immune-mediated demyelination in which the immunomodulatory role of Treg has been widely studied, while our focus is to understand how ageing impairs Treg CNS regenerative roles. In terms of blocking ITGA2 or MCAM *in vivo* and addressing remyelination, we have not performed this experiment because

MCAM and ITGA2 are expressed by a wide range of cells (e.g. astrocytes) and as such, blocking MCAM or ITGA2 *in vivo* could a) yield unexpected off-target effects and b) change remyelination in a non-Treg dependent manner, which would not further our understanding of the role of these two proteins in Treg-mediated remyelination.

4. Little information is provided regarding the mechanism through which aged Tregs have a reinvigorated ability to promote OPC maturation and remyelination following transfer into lysolecithin lesioned young mice. For example, does transfer of young Tregs into aged Treg-depleted and lesioned mice result in a loss of OPC maturation, etc? Does this reinvigoration mechanism require cell-cell contact? It seems this could be easily worked out using their cerebellar slice co-cultures.

To address this, we performed the same Treg adoptive transfer experiments in aged Foxp3DTR mice depleted of Treg (**Fig. 5**). Our results show that transfer of young or aged Treg failed to rescue the decreased Olig2⁺ cell density in the aged spinal cord (**Fig. 5B-D**). However, it appears that young Treg more efficiently increase OPC differentiation in the aged animals than the aged Treg (as shown by Olig2+CC1⁺ cell density), although this increase was not a full rescue (**Fig. 5F, G**). Nevertheless, as indicated in the response to reviewer 1, we could not detect Tregs in the classic lymphoid organs (spleen and lymph node) of aged mice (**Sup. Fig. 4E, F**), and thus, we are cautious about the interpretation of the results.

RFP⁺ aged Treg adoptively transferred into young Treg-depleted mice and then isolated from spleen demonstrated increased expression of MCAM and ITGA2 which suggests that in principle, Treg do not need to be in contact with OPCs to boost the expression of these molecules.

Reviewer #3 (Remarks to the Author):

This manuscript is about the role of T-reg in CNS myelin repair. It has been shown previously that T-reg from young animals can promote remyelination. Here, the team is studying the impact of aging on this specific T-reg function. The topic is timely and exciting; however, the team needs to address issues to strengthen the results.

We thank the reviewer for his/her positive comments regarding the excitement and timeliness of our findings and address his/her concerns point by point below.

The main concern is related to the location of the T-reg, considering that contact is required to promote differentiation *in vitro* (Figure 5). The team must show that they are in the CNS in close proximity to OPCs to promote remyelination.

As we reported previously, numbers of Treg in lysolecithin-induced lesions are very low (1-2 cells/mg of tissue by flow cytometry, Dombrowski *et al.* 2017, Nat Neuro), limiting the number of Treg that can be studied in a spinal cord lesion. Thus, to address this question, we looked for evidence of Treg and OPC interacting by staining demyelinated areas in the murine spinal cord tissue from mice with EAE for CD4, Foxp3 and Olig2 and observed Treg in close proximity to Olig2⁺ cells (**Fig. 7B**).

Figure 1: the team shows that cells expansion in old mice does not promote myelin repair using the lysolecithin model. The team should show baseline count for oligodendrocyte (panel E and I) as age probably impact these numbers.

As suggested by the reviewer we now provide the baseline count for oligodendrocyte lineage cells in the normal appearing white matter (NAWM) of these animals, in which no lesion is present. No baseline differences were observed in the density of Olig2⁺ oligodendrocyte lineage cells, Olig2⁺CC1⁺ oligodendrocytes or in the percentage of Olig2⁺ cells expressing CC1 between young and age normal appearing white matter (**Sup. Fig. 2A-D**). This aligns with single cell RNA sequencing findings (Ximerakis et al, Nat Neuro, 2019; Allen et al., Cell, 2022), that reported no changes in the proportions of oligodendrocytes with age.

The authors should present the data showing that expansion of T-reg in young animals promote remyelination to allow the readers to understand the data better; this piece is critical since authors shows in figure 3 that transfer of old T-reg in a young mice promote myelin repair.

To address this, we examined oligodendrocyte differentiation in remyelinating lesions of young mice following Treg expansion mice at 7 days post lesion (dpl). We chose this earlier timepoint as our previous data at 14dpl showed no difference in oligodendrocyte differentiation, but our previous publication demonstrated that Treg *accelerate* oligodendrocyte differentiation and remyelination (Dombrowski *et al.* 2017). Our new findings show that, indeed, Treg expansion in young mice accelerated the remyelination response as demonstrated by increased oligodendrocyte differentiation (Olig2⁺CC1⁺) at 7dpl (**Fig. 2 D-F**) but this did not occur in aged mice either at 7dpl or at 10dpl (to account for slower remyelination in aged mice) (**Fig. 2 G-I**).

Further the authors show expansion of T-reg in the periphery but what about the CNS (meninges and parenchyma).

We have now quantified Treg in the CNS parenchyma of Treg-expanded young and aged mice by IHC and observe a trend of higher densities of Treg in the Treg-expanded mice (**Sup Fig. 2 G-J**), however this is not statistically significant. This may be due to the inherently low number of Treg in the CNS which, even when expanded locally in the CNS, remain difficult to detect (Yshii et al., Nat Immunology, 2022).

Figure 2: Line 84 and the paragraph need to be edited, authors only look at MBP expression in vitro and myelin development in slices – not myelin regeneration!

We thank the reviewer for pointing out this mistake. We have changed the paragraph accordingly.

Figure 3 to 5: Does the transcriptome of young and old T-reg injected in T-reg depleted young animals equalize. Showing no difference in expression of MCAM and ITGA2 by qpcr and or flow cytometry would prove that the in vitro mechanism is possible in vivo

As suggested by the reviewer, we adoptively transferred young and aged Foxp3-RFP⁺ Treg into young Treg-depleted mice. We recovered RFP⁺ cells by FACS 11 days later and measured MCAM and ITGA2 protein expression by flow cytometry (**Sup Fig. 6**). We detected a decrease in the median fluorescent intensity of those RFP⁺ITGA2⁺ or RFP⁺MCAM⁺ aged cells when compared to young cells. Interestingly, the process of adoptive transfer boosted ITGA2 expression, and to a lesser extent MCAM expression,

in aged cells confirming that aged Treg have capacity to increase MCAM and ITGA2 expression in a young environment *in vivo*. However, due to experimental limitations (e.g problems with faulty isolation kits) and the limited availability of aged-transgenic Foxp3-RFP mice, we have a very low number in this experiment (n=2-3) which is why we did not perform statistical analysis on this result and as mentioned above, has been included as supplementary data.

While the authors indicate the number of mice used for each panel (n), they do not include the number of independent replicate (N). Each key result should have an N of 2.

We have clarified these data in the figure legend including the number of independent mice and independent replicates.

REVIEWERS' COMMENTS

Reviewer #1 (Remarks to the Author):

The authors addressed many of the comments raised by the reviewers.

Further minor comments:

1. Please summarize the current knowledge regarding the presence and frequency of Tregs in MS lesions.
2. In the discussion the author changed the sentence "This is partly due to impaired remyelination with age which was first described in animal models of remyelination (9,10)" to "This is partly due to impaired remyelination with age, which was first described in animal models of remyelination (9,10) but has now also been observed in MS patients (8,11)" However, the references 8 and 11 do not support this new statement. Reference 8 describes (based on a cohort of 20 patients with a maximal observation period of 4 months) a relationship between remyelination and tissue integrity in surrounding white matter, reference 11 reports a better treatment response to a certain remyelination promoting drug in younger compared to older patients. At least to the knowledge of this reviewer there is still no study published, which analyzes spontaneous remyelination in young adult compared to older MS patients. There is one imaging study, describing better remyelination in children compared to young adults, but this rather indicates developmental changes and not an age-associated impairment of remyelination. Please modify your statement accordingly.

Reviewer #2 (Remarks to the Author):

I have no further concerns.

Reviewer #3 (Remarks to the Author):

The authors have undertaken thorough efforts to address my comments. They demonstrate diligence in drawing cautious conclusions, particularly when there is a disparity between in vitro and in vivo data. Moreover, they provide reasoned explanations for instances where experiments may not have been successful. Despite these minor weaknesses, the paper is truly captivating and constitutes a valuable addition to the literature on remyelination and aging.

De la Fuente et al. **Ageing impairs the regenerative capacity of regulatory T cells in central nervous system remyelination**

RESPONSE TO REVIEWERS' COMMENTS

Reviewer #1 (Remarks to the Author):

The authors addressed many of the comments raised by the reviewers.

Further minor comments:

1. Please summarize the current knowledge regarding the presence and frequency of Tregs in MS lesions.

Treg are found in limited numbers across all types of MS lesions with higher numbers in active lesions and being nearly absent in samples from progressive MS patients as well as in lymphoid aggregate structures associated with MS severity. We have included this in the introduction (Line 76-77).

D

2. In the discussion the author changed the sentence "This is partly due to impaired remyelination with age which was first described in animal models of remyelination (9,10)" to "This is partly due to impaired remyelination with age, which was first described in animal models of remyelination (9,10) but has now also been observed in MS patients (8,11)" However, the references 8 and 11 do not support this new statement. Reference 8 describes (based on a cohort of 20 patients with a maximal observation period of 4 months) a relationship between remyelination and tissue integrity in surrounding white matter, reference 11 reports a better treatment response to a certain remyelination promoting drug in younger compared to older patients. At least to the knowledge of this reviewer there is still no study published, which analyzes spontaneous remyelination in young adult compared to older MS patients. There is one imaging study, describing better remyelination in children compared to young adults, but this rather indicates developmental changes and not an age-associated impairment of remyelination. Please modify your statement accordingly.

We thank the reviewer for highlight this ambiguity and we have modified the sentence accordingly to reflect reviewer's input (Lines 290-296).

Reviewer #2 (Remarks to the Author):

I have no further concerns.

Reviewer #3 (Remarks to the Author):

The authors have undertaken thorough efforts to address my comments. They demonstrate diligence in drawing cautious conclusions, particularly when there is a disparity between in vitro and in vivo data. Moreover, they provide reasoned explanations for instances where experiments may not have been successful. Despite these minor weaknesses, the paper is truly captivating and constitutes a valuable addition to the literature on remyelination and aging.

We highly appreciate reviewer's encouraging comments and appreciation of the extensive work undertaken as well as the relevance of this work for the field of remyelination and aging.